# Synthesis and Cytotoxicity of 7,9-*O*-Linked Macrocyclic C-Seco Taxoids

**DOI:** 10.3390/molecules24112161

**Published:** 2019-06-08

**Authors:** Yu Zhao, Tian-En Wang, Alberto Mills, Federico Gago, Wei-Shuo Fang

**Affiliations:** 1State Key Laboratory of Bioactive Substances and Functions of Natural Medicines, Institute of Materia Medica, Chinese Academy of Medical Sciences and Peking Union Medical College, 2A Nan Wei Road, Beijing 100050, China; zhaoyu8410@126.com (Y.Z.); wangtianen@imm.ac.cn (T.-E.W.); 2Department of Biomedical Sciences and “Unidad Asociada IQM-CSIC”, School of Medicine and Health Sciences, University of Alcalá, E-28805 Alcalá de Henares, Madrid, Spain; albertomillsfernandez@gmail.com (A.M.); federico.gago@uah.es (F.G.)

**Keywords:** taxoids, βIII-tubulin, P-glycoprotein, drug resistance

## Abstract

A series of novel 7,9-*O*-linked macrocyclic taxoids together with modification at the C2 position were synthesized, and their cytotoxicities against drug-sensitive and P-glycoprotein and βIII-tubulin overexpressed drug-resistant cancer cell lines were evaluated. It is demonstrated that C-seco taxoids conformationally constrained via carbonate containing-linked macrocyclization display increased cytotoxicity on drug-resistant tumors overexpressing both βIII and P-gp, among which compound **22b**, bearing a 2-m-methoxybenzoyl group together with a five-atom linker, was identified as the most potent. Molecular modeling suggested the improved cytotoxicity of **22b** results from enhanced favorable interactions with the T7 loop region of βIII.

## 1. Introduction

Antitubulin agent paclitaxel (**1a**, Figure 1) and its semi-synthetic derivative docetaxel (**1b**, Figure 1) are successfully used in the clinic for the treatment of ovarian, breast, and non-small cell lung and prostate cancers and Kaposi’s sarcoma. 

However, their clinical uses are severely restricted by drug resistance. Various resistance mechanisms toward taxane-based anticancer drugs have been revealed, among which only overexpression of P-glycoprotein (P-gp) and βIII-tubulin have been confirmed clinically [1,2,3,4]. 

A great deal of effort has been made to overcome multi-drug resistance (MDR) mediated by overexpression of P-gp. Nevertheless, many taxane-based drug candidates targeting P-gp overexpression did not achieve the expected efficacy in clinical trials [5,6]. 

A correlation between βIII-tubulin overexpression and poor prognosis has been reported in several human tumors, including various advanced malignancies upon treatment of taxanes, such as breast, non-small cell lung, and ovarian cancers [7,8,9,10]. It was reported that a C-seco taxane IDN5390 (**2**, Figure 1) is more active than paclitaxel in several paclitaxel-resistant human ovarian adenocarcinoma cell lines (e.g., A2780TC1, A2780TC3, and OVCAR-3), expressing high βIII-tubulin and P-gp levels. However, IDN5390 was nearly 10-fold less active than paclitaxel in paclitaxel-sensitive cells [11]. 

To find a modified taxane effective against both paclitaxel-sensitive and -resistant tumors, the 7,9-*O*-linked C-seco taxoid **3** (Figure 2) was synthesized in our lab and shown to exhibit significant activity enhancement against drug-sensitive Hela and βIII-tubulin overexpressing HeLa-βIII tumor cells compared to the C-seco paclitaxel derivative **4** (an analog of IDN5390, Figure 2) [12]. Those macrocyclic taxoids provided a successful example for the compromise between structural pre-organization and sufficient flexibility in the C ring of C-seco taxoids to improve the cytotoxicity against tumor cells that overexpress βIII tubulin [12]. 

It has long been observed that modification of taxane at the C2 position could play a critical role in its interaction with tubulin. Consistent with this observation, the 7,9-*O*-linked C-seco taxoid with C2 modification **5** (Figure 3) was much more cytotoxic than the C-seco taxoid **6** (Figure 3) in drug-resistant A2780AD and KB-V1 cells (P-gp overexpression) and Hela-βIII (βIII overexpression) [12]. It was demonstrated that the *meta*-substitution (i.e., -OMe, -F, -Cl) of the C2-benzoate moiety of C-seco taxoids derivatives **7a**–**c** (Figure 3) could increase the interaction of C-seco-taxoids with βIII-tubulin to overcome paclitaxel-resistance [13]. 

In order to explore the effect of macrocyclization in a different way from that employed in our previous study (i.e., via the carbonate formation to restrict the conformation), the synthesis and biological activity assessment of a series of 7,9-*O*-linked macrocyclic taxoids together with modification at the C2 position were performed and are reported here. 

## 2. Results and Discussion

### 2.1. Chemistry

Our synthesis began with the preparation of 2′-*tert*-butyldimethylsilyl-paclitaxel **12a**, which was then converted to 10-deacetylpaclitaxel **13a** upon treatment with hydrazine hydrate in methanol. Synthesis of the analogues **12b**–**e** of 2′-*tert*-butyldimethylsilyl-paclitaxel with modifications at the C2 positions was then realized by selective C2 debenzoylation and then elaboration with various *m*-substituted benzoyl group. C2-modified 10-deacetylpaclitaxel **13b**–**e** were afforded in the same manner as compound **13a**. Then the compounds **13a**–**e** were oxidized by copper (II) diacetate to furnish **14a**–**e** as a mixture of two epimers. Reductive trapping of the ring C-seco tautomers **15a**–**e** were obtained by treatment with l-selectride. Subsequent deprotection with pyridine hydrofluoride (HF-pyridine) afforded **16a**–**e**. The 9-OH in **15a**–**e** was hydroxylalkylated by 2-bromoethanol or 3-bromo-1-propanol in the presence of potassium carbonate and potassium iodide in *N*,*N*-dimethylformamide (DMF). Cyclization of compounds **17a**–**e** could be accomplished by treatment with a slight excess of triphosgene, affording the cyclic carbonate analogues **19a**–**e**. Desilylation of **19a**–**e** with HF-pyridine afforded macrocyclic analogues **21a**–**e**. Accordingly, macrocyclic taxoids **22a**–**e** bearing a linkage with one more carbon atom were synthesized from **18a**–**e** in a similar manner (Scheme 1).

### 2.2. Bioactivity Evaluation of Taxoids ***21a**–**e*** and ***22a**–**e***

Taxoids **21a**–**e**, **22a**–**e**, and **16a**–**e** were evaluated for their in vitro cytotoxicities against cervical carcinoma HeLa cells and their corresponding drug-resistant cell lines (HeLa-βIII). 

As shown in Table 1, all the 7,9-*O*-linked macrocyclic analogues (**21a**–**e, 22a**–**e**) possessed a remarkably higher potency against HeLa and HeLa-βIII cells than their corresponding C-seco taxoids analogues **16a**–**e**. The R/S values (IC_50_ in drug resistant cells/IC_50_ in drug sensitive cells) of all macrocyclic analogues were lower than those of their corresponding C-seco taxoids analogues **16a**–**e**. These findings demonstrate that conformational restraint via carbonate-containing linked macrocyclization can improve the cytotoxicity against human carcinoma cell lines overexpressing βIII tubulin. 

Furthermore, all of the 7,9-*O*-linked macrocyclic analogues bearing the four-atom linker (**21a**–**e**) turned out to be generally more potent against sensitive HeLa cells and HeLa-βIII cells than the corresponding macrocyclic analogues bearing the five-atom linker (**22a**–**e**). Since the cytotoxicity of **21a**–**d** was approximately equal to that of paclitaxel, these results suggest that a four-atom tether is optimal in these macrocyclic C-seco taxoids. 

For the effect of C2 modifications in these macrocyclic C-seco taxoids, we found similar results as previously reported [12,13]. Pepe et al. studied the cytotoxicity of a series of C-seco taxoids, whereby the introduction of a substituent (-OMe, -F, -Cl) to the *meta*-position of the C2 benzoyl moiety increased potency against the βIII-tubulin-overexpressing, paclitaxel-resistant cell line A2780TC3 [13]. Taxoid **6** bearing a 2-(*m*-azido) benzyloxy group and synthesized in our lab exhibited higher binding affinity for microtubulin (MT) than its corresponding taxoids bearing a 2-benzyloxy group [12]. The 7,9-*O*-linked C-seco taxoid bearing a 2-(*m*-azido) benzyloxy group **5** was significantly more active than **6** on drug-resistant A2780AD, HeLa-βIII, and KB-V1 cells [12]. 

We investigated this effect in the 7,9-*O*-linked C-seco taxoids **21b**–**e** and **22b**–**e** reported here. Thus, introduction of a *meta* substituent (-OMe, -F, -Cl) on derivatives bearing the four-atom linker showed similar cytotoxicity against HeLa sensitive cell lines and Hela-βIII cells lines as their corresponding C-seco taxoid **21a** bearing a 2-benzyloxy group. In contrast, the *meta*-substituted (-OMe, -F, -Cl) derivatives bearing five-atom linker possessed higher potency against HeLa-βIII cells lines than its corresponding C-seco taxoid **22a** (by a factor of 3.1–8.9). Nonetheless, the 2-*m*-CF_3_ analogues **21e** and **22e** were considerably less cytotoxic against both drug sensitive and resistant cells by one to two orders of magnitude.

The growth inhibition effects of the 7,9-*O*-linked macrocyclic analogues were also measured on human breast cancer MCF-7 cells and their corresponding P-gp-overexpressing drug-resistant MCF-7/R cells. As shown in Table 1, the 7,9-*O*-linked macrocyclic analogues possessed remarkable potency against MCF-7/R cells compared to their C-seco counterparts, except for the 2-*m*-CF_3_ derivatives. Strikingly, macrocyclic taxoids bearing the five-atom linker were almost inactive. The cytotoxicity against MCF-7/R cells of the 7,9-*O*-linked macrocyclic derivatives (-OMe, -F, -Cl) bearing the five-atom linker were slightly higher than that of the compouds bearing the four-atom linker. Compounds **22b**–**d** (-OMe, -F, -Cl) bearing the five-atom linker and **21a**, **21d** (-H, -Cl) bearing the four-atom linker displayed lower R/S values than that of paclitaxel. All in all, **22b** turned out to be the best compound in the whole series.

### 2.3. Binding Mode of ***22b*** and Rationale for the Increased Affinity for the βIII Isotype

Inspection of the paclitaxel-β-tubulin complex in high-resolution cryo-electron microscopy (cryo-EM) structures of mammalian microtubules reveals a very tight fit between the bulky drug and the taxane-binding site, in which the M-loop is structured as an α-helix [14]. The oxetane ring oxygen of paclitaxel (**1a**) accepts a hydrogen bond from the backbone NH of Thr276, whereas the C-3′ benzamide carbonyl oxygen acts as a hydrogen bond acceptor for N^ε^ of His229. These two hydrogen bonds are likely to be present in the **22b**:β-tubulin complex as well (Figure 4), and definitely contribute to the binding affinity, but these two residues are common to both βIIB and βIII isotypes. In the search for sequence differences that can account for the resistance to taxanes, a lot of emphasis has been placed on the Ser→Ala replacement at position 277, but we believe that the most important replacement in relation to drug resistance possibly involves Cys241→Ser. The reason is that, as a consequence of this substitution, disulfide bond formation with Cys356 is precluded in the βIII isotype and the T7 loop region, which closes over the bound taxane, is likely to be more flexible. By directing the macrocyclic region of **22b** towards this loop, favorable interactions within the taxane-binding site are enhanced and the affinity towards the βIII isotype is improved. We hypothesize that the improved cytotoxicity of **22b** relative to **1a**, and the other taxanes described here, results from enhanced favorable interactions with the T7 loop region of βIII, an isotype in which Cys356 cannot engage in a disulfide bond with the amino acid present at position 241 (Ser).

As shown in this modelling study, the binding of these taxoids to microtubules are similar to that of paclitaxel. Although cyclization does increase the cytotoxicity either in drug-sensitive and -resistant cells in most cases, the 7,9-*O*-linked macrocyclic taxoids only showed comparable cytotoxicity to that of paclitaxel. The enhanced activity is not only arisen from the known C-2 modifications [15,16], but also from the cyclization of the cleaved C-ring of taxane.

## 3. Experimental

### 3.1. Chemical Synthesis

#### 3.1.1. General Methods

All chemicals and reagents were purchased from Beijing Innochem Science and Technology Co. Ltd. (Beijing, China), Sinopharm Chemical Reagent Co. Ltd. (Beijing, China), and thin-layer chromatography (TCI). The 200–300 mesh silica gel used for flash column chromatography was purchased from Rushanshi Shuangbang Xincailiao Co. Ltd. (Rushan, Shandong, China). Visualization on TLC (analytical thin layer chromatography) was achieved by the use of UV light (254 nm) and treatment with phosphomolybdic acid or KMnO_4_ followed by heating. All solvents were purified and dried according to the standard procedures. The purification was performed on flash column chromatography. The high performance liquid chromatography (HPLC)- electrospray ionization (ESI)-mass spectrometry (MS) analysis was carried out in an Agilent 1260 Infinity HPLC system (Agilent Technologies, Waldbronn, Germany) equipped with a reversed phase 4.68 × 50 mm (1.8 um) XDB-C18 Column and consisted of a binary solvent delivery system, an auto sampler, a column temperature controller, and an UV detector. The mass spectra were acquired by a 6120 Quadrupole LC-MS mass spectrometer (Agilent Technologies, Waldbronn, Germany) connected to the HPLC system via an ESI interface. Proton and carbon magnetic resonance spectra (^1^H-NMR and ^13^C-NMR) were recorded on a Bruker BioSpin AG 300 or 400 MHz spectrometer or Varian 300, 400, or 600 MHz spectrometer.^1^H-NMR data were reported as follows: Chemical shifts, multiplicity (s = singlet, d = doublet, t = triplet, q = quartet, m=multiplet), coupling constant(s) in Hz, integration. All tested compounds **16a**–**e**, **21a**–**e**, and **22a**–**e** were ≥95% pure by HPLC (column XDB C18 4.6 × 50 mm 1.8 μm, mobile phase: acetonitrile-water (10:90–100:0 gradient in 4.5 min), flow rate 1.0 mL/min, detected at 220 nm). 

#### 3.1.2. General Experimental Procedure for Compounds **14a**–**e**

*2′-O-(tert-Butyldimethylsilyl)-10-dehydro-10-deacetylpaclitaxel* (**14a**). To a solution of **13a** (261.3 mg, 0.282 mmol) in methanol (8 mL) was added Cu(OAc)_2_·H_2_O (282 mg, 1.41 mmol). The suspension was stirred at room temperature overnight (open air) and then diluted with ethyl acetate. The organic phase was washed sequentially with saturated aqueous NaCl (50 mL), 2 N NH_3_ to remove copper salts and saturated aqueous NH_4_Cl. After drying (Na_2_SO_4_) and evaporation under reduced pressure, the crude product was purified by silica gel chromatography (ethyl acetate: petroleum ether = 1:2) to give a 4:1 mixture of epimers **14a** (14-β-OH and 14-α-OH, 223.8 mg, 85.8% total yield). ^1^H-NMR (CDCl_3_, 600 MHz) δ: 8.19 (d, *J* = 7.8 Hz, 1H, Ar-H), 7.73 (d, *J* = 7.8 Hz, 2H, Ar-H), 7.48 (m, 11H, Ar-H), 7.06 (d, *J* = 9.0 Hz, 1H, NH), 6.28 (t, *J* = 8.7 Hz, 1H, H-13), 5.87 (d, *J* = 7.2 Hz, 1H, H-2), 5.79 (dd, *J* = 1.5, 8.7 Hz, 1H, H-3′), 4.93 (m, 1H, H-5), 4.67 (d, *J* = 1.8 Hz, 1H, H-2′), 4.50 (d, *J* = 11.4 Hz, 1H, H-7), 4.44 (d, *J* = 8.4 Hz, 1H, H- 20a), 4.38 (d, *J* = 9.0 Hz, 1H, H-20b), 4.04 (d, *J* = 7.2 Hz, 1H, H-3), 3.88 (m, 1H, OH-7), 2.67 (s, 3H, CH_3_), 2.48 (dd, *J* = 9.6, 15.6 Hz, 1H, H-6a), 2.29 (m, 2H, H-14b, H-14a), 2.19 (m, 1H, H-6b), 1.86 (s, 3H, CH_3_), 1.75 (s, 3H, CH_3_), 1.21 (s, 3H, CH_3_), 1.13 (s, 3H, CH_3_), 0.79 (s, 9H, SiC(CH_3_)_3_), −0.04 (s, 3H, Si(CH_3_)), −0.30 (s, 3H, Si(CH_3_)). (For the chemical characterization refer to http://dx.doi.org/10.1016/j.ejmech.2017.06.001)

*2′-O-(tert-Butyldimethylsilyl)-2-debenzoyl-2-(3-methoxybenzoyl)-7-epi-10-dehydro-10-deacetylpaclitaxel* (**14b**). 69.5% yield as a 4:1 mixture of two epimers; ^1^H-NMR (CDCl_3_, 400 MHz) δ: 7.80 (d, *J* = 7.7 Hz, 1H, Ar-H), 7.74 (d, *J* = 7.2 Hz, 2H, Ar-H), 7.41 (m, 10H, Ar-H), 7.21 (dd, *J* = 2.0, 8.0 Hz, Ar-H), 7.09 (d, *J* = 9.1 Hz, 1H, NH), 6.27 (t, *J* = 8.9 Hz, 1H, H-13), 5.89 (d, *J* = 7.2 Hz, 1H, H-2), 5.79 (dd, *J* = 2.2, 9.4 Hz, 1H, H-3′), 4.94 (m, 1H, H-5), 4.67 (d, *J* = 2.0 Hz, 1H, H-2′), 4.50 (d, *J* = 11.2 Hz, 1H, H-7), 4.47 (d, *J* = 8.8 Hz, 1H, H- 20a), 4.38 (d, *J* = 8.6 Hz, 1H, H-20b), 4.05 (d, *J* = 7.1 Hz, 1H, H-3), 3.90 (s, 3H, OCH_3_), 3.88 (m, 1H, OH-7), 2.67 (s, 3H, CH_3_), 2.49 (dd, *J* = 9.8, 15.4 Hz, 1H, H-6a), 2.29 (m, 2H, H-14b, H-14a), 1.86 (s, 3H, CH_3_), 2.25 (m, 1H, H-6b), 1.75 (s, 3H, CH_3_), 1.21(s, 3H, CH_3_), 1.13 (s, 3H, CH_3_), 0.81 (s, 9H, SiC(CH_3_)_3_), −0.02 (s, 3H, Si(CH_3_)), −0.26 (s, 3H, Si(CH_3_)); ^13^C-NMR (CDCl_3_, 100 MHz) δ: 207.8, 188.7, 172.2, 170.9, 167.0, 166.9, 159.8, 143.3, 141.1, 138.2, 134.1, 131.9, 130.3, 130.0, 128.8 (2C), 128.1, 127.0, 126.4, 122.7, 120.5, 114.4, 82.7, 81.8, 79.2, 77.4, 75.3, 75.1, 70.7, 57.2, 55.5 (2C), 40.3, 39.4, 36.3, 35.4, 26.9, 26.1, 25.5, 22.9, 18.2, 15.0, 14.4, −5.2, −5.8; LC-MS (ESI, *m/z*): [M + 1], found 954.4, [M + 23], found 976.4, C_52_H_63_NO_14_Si: 953.40 (see supplementary materials).

*2′-O-(tert-Butyldimethylsilyl)-2-debenzoyl-2-(3-fluorobenzoyl)-7-epi-10-dehydro-10-deacetylpaclitaxel* (**14c**). 79.0% yield as a 4:1 mixture of two epimers; ^1^H-NMR (CDCl_3_, 400 MHz) δ: 8.02 (d, *J* = 8.0 Hz, 1H, Ar-H), 7.89–7.86 (m, 1H, Ar-H), 7.73 (d, *J* = 7.2 Hz, 2H, Ar-H), 7.59–7.33 (m, 10H, Ar-H), 7.07 (d, *J* = 9.2 Hz, 1H, BzNH), 6.27 (t, *J* = 8.8 Hz, 1H, H-13), 5.86 (d, *J* = 7.2 Hz, 1H, H-2), 5.80 (dd, *J* = 2.0, 9.2 Hz, 1H, H-3′), 4.95 (dd, *J* = 4.0, 8.0 Hz, 1H, H-5), 4.68 (d, *J* = 2.4 Hz, 1H, H-2′), 4.48 (d, *J* = 11.2 Hz, 1H, H-7), 4.44 (d, *J* = 8.8 Hz, 1H, H- 20a), 4.36 (d, *J* = 8.8 Hz, 1H, H-20b), 4.05 (d, *J* = 7.2 Hz, 1H, H-3), 3.92–3.88 (m, 1H, OH-7), 2.68 (s, 3H, COCH_3_), 2.48 (dd, *J* = 9.6, 15.2 Hz, 1H, H-6a), 2.31–2.29 (m, 2H, H-14b, H-14a), 2.22 (dd, *J* = 8.4, 15.6 Hz, 1H, H-6b), 1.99 (br, 1H, OH), 1.86 (s, 3H, CH_3_), 1.76 (s, 3H, CH_3_), 1.22 (s, 3H, CH_3_), 1.13 (s, 3H, CH_3_), 0.81 (s, 9H, SiC(CH_3_)), −0.02 (s, 3H, Si(CH_3_)), −0.26 (s, 3H, Si(CH_3_)_2_); ^13^C-NMR (CDCl_3_, 100 MHz) δ: 207.8, 188.6, 172.3, 171.2, 170.9, 167.0, 165.9 (d, ^4^*J*_C-F_ = 2.9 Hz), 162.7 (d, ^1^*J*_C-F_ = 246.5 Hz), 143.5, 140.9, 138.2, 134.1, 131.9, 131.3 (d, ^3^*J*_C-F_ = 7.3 Hz), 130.7 (d, ^3^*J*_C-F_ = 7.6 Hz), 128.83 (2C), 128.77 (2C), 128.1, 127.0 (2C), 126.4 (2C), 126.1 (d, ^4^*J*_C-F_ = 3.1 Hz), 120.0 (d, ^2^*J*_C-F_ = 21.1 Hz), 117.1 (d, ^2^*J*_C-F_ = 23.1 Hz), 82.8, 81.7, 79.4, 75.5, 75.3, 70.6, 57.2, 55.6, 40.3, 39.5, 36.3, 35.4, 26.1, 25.5 (3C), 22.9 (2C), 18.2, 15.0, 14.5, 14.2, −5.2, −5.8; LC-MS (ESI, *m/z*): [M + 1], found 942.4, [M + 23], found 964.3, C_51_H_60_FNO_13_Si: 941.38. 

*2′-O-(tert-Butyldimethylsilyl)-2-debenzoyl-2-(3-chlorobenzoyl)-7-epi-10-dehydro-10-deacetylpaclitaxel* (**14d**). 84.9% yield as a 4:1 mixture of two epimers; ^1^H-NMR (CDCl_3_, 400 MHz) δ: 8.18 (s, 1H, Ar-H), 8.11 (d, *J* = 7.6 Hz, 1H, Ar-H), 7.73 (d, *J* = 7.6 Hz, 2H, Ar-H), 7.62 (d, *J* = 7.6 Hz, 1H, Ar-H), 7.55–7.50 (m, 2H, Ar-H), 7.44–7.35 (m, 7H, Ar-H), 7.08 (d, *J* = 8.8 Hz, 1H, BzNH), 6.25 (t, *J* = 8.8 Hz, 1H, H-13), 5.84 (d, *J* = 7.2 Hz, 1H, H-2), 5.79 (dd, *J* = 8.8 Hz, 1H, H-3′), 4.96 (m, 1H, H-5), 4.67 (d, *J* = 2.0 Hz, 1H, H-2′), 4.47 (d, *J* = 11.2 Hz, 1H, H-7), 4.44 (d, *J* = 8.4 Hz, 1H, H- 20a), 4.35 (d, *J* = 8.4 Hz, 1H, H-20b), 4.05 (d, *J* = 7.2 Hz, 1H, H-3), 3.89 (m, 1H, OH-7), 2.68 (s, 3H, CH_3_), 2.48 (dd, *J* = 10.0, 15.4 Hz, 1H, H-6a), 2.29 (m, 2H, H-14b, H-14a), 2.23 (m, 1H, H-6b), 1.86 (s, 3H, CH_3_), 1.76 (s, 3H, CH_3_), 1.22 (s, 3H, CH_3_), 1.13 (s, 3H, CH_3_), 0.82 (s, 9H, SiC(CH_3_)_3_), −0.02 (s, 3H, Si(CH_3_)), −0.26 (s, 3H, Si(CH_3_)); ^13^C-NMR (CDCl_3_, 100 MHz) δ: 207.8, 188.6, 172.2, 170.9, 165.7, 143.5, 140.9, 138.2, 134.9, 134.1, 133.9, 131.9, 130.9, 130.4, 130.3, 128.83 (2C), 128.77 (2C), 128.5, 128.1, 127.0 (2C), 126.5 (2C), 82.7, 81.7, 79.4, 75.4, 75.3, 70.7, 60.4, 57.2, 55.6, 40.2, 39.4, 36.3, 35.3, 29.7, 26.1, 25.5 (2C), 22.9, 22.8, 18.2, 14.9, 14.5, −5.2, −5.8; LC-MS (ESI, *m/z*): [M + 1], found 958.4, [M + 23], found 980.4, C_51_H_60_ClNO_13_Si: 957.35.

*2′-O-(tert-Butyldimethylsilyl)-2-debenzoyl-2-(3-trifluoromethyl)-7-epi-10-dehydro-10-deacetylpaclitaxel* (**14e**). 81.9% yield as a 4:1 mixture of two epimers; ^1^H-NMR (CDCl_3_, 400 MHz) δ: 8.50 (s, 1H, Ar-H), 8.43 (d, *J* = 7.6 Hz, 1H, Ar-H), 7.91 (d, *J* = 8.0 Hz, 1H, Ar-H), 7.72 (d, *J* = 7.8 Hz, 3H, Ar-H), 7.51 (m, 1H, Ar-H), 7.41 (m, 7H, Ar-H), 7.06 (d, *J* = 9.2 Hz, 1H, BzNH), 6.24 (t, *J* = 8.6 Hz, 1H, H-13), 5.88 (d, *J* = 7.6 Hz, 1H, H-2), 5.78 (dd, *J* = 1.6, 9.2 Hz, 1H, H-3′), 4.97 (m, 1H, H-5), 4.67 (d, *J* = 2.0 Hz, 1H, H-2′), 4.45 (d, *J* = 11.2 Hz, 1H, H-7), 4.42 (d, *J* = 8.5 Hz, 1H, H-20a), 4.37 (d, *J* = 8.6 Hz, 1H, H-20b), 4.09 (d, *J* = 7.6 Hz, 1H, H-3), 3.90 (m, 1H, OH-7), 2.66 (s, 3H, CH_3_), 2.51 (dd, *J* = 9.6, 15.5 Hz, 1H, H-6a), 2.30 (m, 2H, H-14b, H-14a), 2.26 (m, 1H, H-6b), 1.87 (s, 3H, CH_3_), 1.77 (s, 3H, CH_3_), 1.23 (s, 3H, CH_3_), 1.15 (s, 3H, CH_3_), 0.83 (s, 9H, SiC(CH_3_)_3_), −0.02 (s, 3H, Si(CH_3_)), −0.24 (s, 3H, Si(CH_3_)); ^13^C-NMR (CDCl_3_, 100 MHz) δ: 207.7, 188.6, 172.1, 171.0, 167.1, 165.6, 143.6, 140.9, 138.3, 134.2, 133.6, 131.9, 131.4 (q, ^2^*J*_C-F_ = 33.2 Hz), 130.3 (q, ^3^*J* = 3.4 Hz), 130.3, 130.1, 129.9, 128.9, 128.8, 128.1, 127.1 (q, ^3^*J*_C-F_ = 4.0 Hz), 127.0, 126.5, 123.7 (q, ^1^*J*_C-F_ = 273.6 Hz), 82.8, 81.7, 79.5, 75.6, 75.3, 70.7, 57.2, 55.6, 40.3, 39.5, 36.3, 35.4, 26.1, 25.6, 22.9, 22.7, 18.2, 14.9, 14.5, −5.2, −5.8; LC-MS (ESI, *m/z*): [M + 1], found 992.0, C_52_H_60_F_3_NO_13_Si: 991.38.

#### 3.1.3. General Experimental Procedure for Compounds **15a**–**e**

*2′-tert-Butyldimethylsilyl-10-dehydro-7-8-seco-10-deacetylpaclitaxel* (**15a**). l-selectride (1M solution in THF, 0.8 mL) was added, dropwise, to a solution of **14a** (171.5 mg, 0.186 mmol) in THF (4.1 mL) at −20 °C. After 15 min, the reaction was quenched by addition of cold ethyl acetate (50.0 mL) and 2 N H_2_SO_4_ (4.0 mL). The reaction mixture was extracted with ethyl acetate and the combined organic phases were washed with a saturated aqueous NaHCO_3_ and saturated aqueous NH_4_Cl. The resulting solution was dried over anhydrous MgSO_4_ and the solvent evaporated under reduced pressure to give a crude product. The crude product was purified by silica gel chromatography (ethyl acetate: petroleum ether = 1:1) which gave product **15a** (139.5 mg, 81.2% yield) as a white solid

*2′-tert-Butyldimethylsilyl-2-debenzoyl-2-(3-methoxybenzoyl)-10-dehydro-7,8-seco-10-deacetylpaclitaxel* (**15b**). White solid; yield 88.7%; ^1^H-NMR (CDCl_3_, 400 MHz) δ: 7.73–7.70 (m, 3H, Ar-H), 7.55 (s, 1H, Ar-H), 7.42 (m, 10H, Ar-H), 7.14 (dd, *J* = 2.3, 8.2 Hz, 1H, BzNH), 6.47 (s, 1H, OH-9), 6.29 (t, *J* = 8.0 Hz, 1H, H-13), 5.91 (d, *J* = 8.8 Hz, 1H, H-3′), 5.60 (d, *J* = 9.2 Hz, 1H, H-2), 5.24 (brs, 2H, H-5, H-20a), 4.72 (d, *J* = 1.6 Hz, 1H, H-2′), 4.29 (d, *J* = 6.8 Hz, 1H, H-3), 4.20 (brs, 1H, H-7a), 3.96 (brs, 1H, OH), 3.85 (s, 4H, OCH_3_, H-7b), 2.82 (brs, 1H, H-6a), 2.71 (brs, 1H, OH), 2.26 (dd, *J* = 9.6, 15.6 Hz, 2H, H-14a, H-14b), 2.09–2.02 (m, 4H, CH_3_, H-6b), 1.87 (brs, 6H, 2 × CH_3_), 1.28 (s, 3H, CH_3_), 1.10 (s, 3H, CH_3_), 0.79 (s, 9H, SiC(CH_3_)_3_), −0.11 (s, 3H, Si(CH_3_)), −0.28 (s, 3H, Si(CH_3_)); ^13^C-NMR (CDCl_3_, 100 MHz) δ: 191.2, 171.2, 171.1, 168.9, 167.4, 167.2, 159.8, 148.9, 142.1, 138.2, 133.9, 131.9, 130.5, 130.3, 128.7 (4C), 128.0, 127.1 (2C), 126.7 (2C), 124.6, 122.2, 119.5, 115.0, 86.3, 76.1, 75.1, 70.6, 59.2, 55.8, 55.5, 43.1, 36.6, 25.5 (3C), 23.4, 22.2, 21.5, 18.3, 14.8, 14.4, −5.6, −6.0; LC-MS (ESI, *m/z*): [M + 1], found 956.4, [M + 23], found 978.4, C_52_H_65_NO_14_Si: 955.42.

*2′-tert-Butyldimethylsilyl-2-debenzoyl-2-(3-fluorobenzoyl)-10-dehydro-7,8-seco-10-deacetylpaclitaxel* (**15c**). White solid; yield 79.2%; ^1^H-NMR (CDCl_3_, 400 MHz) δ: 7.94 (d, *J* = 7.8 Hz, 1H, Ar-H), 7.76 (d, *J* = 8.7 Hz, 1H, Ar-H), 7.68 (d, *J* = 7.4 Hz, 2H, Ar-H), 7.53–7.44 (m, 3H, Ar-H), 7.40–7.30 (m, 8H, NH, Ar-H), 6.48 (s, 1H, OH-9), 6.30 (t, *J* = 8.0 Hz, 1H, H-13), 5.92 (d, *J* = 9.0 Hz, 1H, H-3′), 5.60 (d, *J* = 9.5 Hz, 1H, H-2), 5.25 (brs, 2H, H-5, H-20a), 4.71 (d, *J* = 2.1 Hz, 1H, H-2′), 4.30 (d, *J* = 7.8 Hz, 1H, H-3), 4.19 (brs, 1H, H-20b), 3.96 (brs, 1H, H-7a), 3.80 (brs, 1H, H-7b), 2.78 (brs, 1H, H-6a), 2.69 (brs, 1H, H-14a), 2.26 (dd, *J* = 9.5, 15.6 Hz, 1H, H-14b), 2.10–2.00 (ovelap, 1H, H-6b), 1.90 (s, 3H, CH_3_), 1.86 (overlap and brs, 6H, 2 × CH_3_), 1.27 (s, 3H, CH_3_), 1.10 (s, 3H, CH_3_), 0.78 (s, 9H, SiC(CH_3_)_3_), −0.09 (s, 3H, Si(CH_3_)), −0.29 (s, 3H, Si(CH_3_)); ^13^C-NMR (CDCl_3_, 100 MHz) δ: 191.1, 171.2, 171.0, 168.8, 167.4, 166.4, 131.0 (d, ^1^*J*_C-F_ = 248.1 Hz), 128.8 (2C), 128.1, 127.1, 126.7, 125.8 (d, ^4^*J*_C-F_ = 2.6 Hz), 124.6, 121.1 (d, ^2^*J*_C-F_ = 21.3 Hz), 116.7 (d, ^2^*J*_C-F_ = 22.6 Hz), 86.5, 80.7, 76.1, 75.5, 70.5, 59.0, 55.8, 44.5, 43.2, 36.7, 25.6, 22.3, 21.6, 18.3, 14.9, 14.5, −5.4, −5.9; LC-MS (ESI, *m/z*): [M + 1], found 944.4, [M + 23], found 966.4, C_51_H_62_FNO_13_Si: 943.40.

*2′-tert-Butyldimethylsilyl-2-debenzoyl-2-(3-chlorobenzoyl)-10-dehydro-7,8-seco-10-deacetylpaclitaxel* (**15d**). White solid; yield 58.8%; ^1^H-NMR (CDCl_3_, 400 MHz) δ: 8.05 (d, *J* = 7.6 Hz, 2H, Ar-H), 8.01 (s, 1H, Ar-H), 7.68 (d, *J* = 7.6 Hz, 2H, Ar-H), 7.57 (d, *J* = 8.0 Hz, 1H, Ar-H), 7.41–7.31 (m, 9H, NH, Ar-H), 6.49 (s, 1H, OH-9), 6.29 (t, *J* = 8.4 Hz, 1H, H-13), 5.92 (d, *J* = 9.3 Hz, 1H, H-3′), 5.59 (d, *J* = 9.5 Hz, 1H, H-2), 5.23 (brs, 2H, H-5, H-20a), 4.71 (d, *J* = 1.6 Hz, 1H, H-2′), 4.30 (d, *J* = 6.9 Hz, 1H, H-3), 4.18 (brs, 1H, H- 20b), 3.98 (brs, 1H, H-7a), 3.83 (brs, 1H, H-7b), 2.78 (brs, 1H, H-6a), 2.67 (brs, 1H, H-14a), 2.25 (dd, *J* = 9.6, 15.6 Hz, 1H, H-14b), 2.05 (m, 1H, H-6b), 1.89 (s, 3H, CH_3_), 1.85 (brs, 6H, 2 × CH_3_), 1.26 (s, 3H, CH_3_), 1.10 (s, 3H, CH_3_), 0.78 (s, 9H, SiC(CH_3_)_3_), −0.11 (s, 3H, Si(CH_3_)), −0.28 (s, 3H, Si(CH_3_)); ^13^C-NMR (CDCl_3_, 100 MHz) δ: 191.1, 171.0, 168.7, 167.4, 166.3, 148.9, 142.0, 138.0, 137.1, 135.0, 133.9, 131.9, 131.0, 130.6, 129.6, 128.7 (2C), 128.1, 128.0, 127.0 (2C), 126.7 (2C), 124.5, 86.4, 87.0, 80.5, 76.0, 75.4, 70.5, 55.7, 43.1, 36.6, 29.7, 26.9, 25.5 (3C), 22.2, 21.5, 18.3, 14.8, 14.5, −5.5, −6.0; LC-MS (ESI, *m/z*): [M + 1], found 960.0, [M + 23], found 981.9, C_51_H_62_ClNO_13_Si: 959.37.

*2′-tert-Butyldimethylsilyl-2-debenzoyl-2-(3-trifluoromethylbenzoyl)-10-dehydro-7,8-seco-10-deacetylpaclitaxel* (**15e**). White solid; yield 76.5%; ^1^H-NMR (CDCl_3_, 400 MHz) δ: 8.38 (d, *J* = 7.9 Hz, 2H, Ar-H), 7.84 (d, *J* = 7.9 Hz, 2H, Ar-H), 7.48 (m, 11H, NH, Ar-H), 6.51 (s, 1H, OH-9), 6.29 (t, *J* = 8.4 Hz, 1H, H-13), 5.89 (d, *J* = 9.3 Hz, 1H, H-3′), 5.62 (d, *J* = 9.4 Hz, 1H, H-2), 5.25 (brs, 2H, H-5, H-20a), 4.72 (brs, 1H, H-2′), 4.32 (d, *J* = 6.8 Hz, 1H, H-3), 4.24 (brs, 1H, H- 20b), 3.94 (brs, 1H, H-7a), 3.76 (brs, 1H, H-7b), 2.78 (brs, 1H, H-6a), 2.66 (brs, 1H, H-14a), 2.28 (dd, *J* = 9.5, 15.6 Hz, 1H, H-14b), 2.05 (m, 1H, H-6b), 1.92 (s, 3H, CH_3_), 1.85 (brs, 6H, 2 × CH_3_), 1.27 (s, 3H, CH_3_), 1.11 (s, 3H, CH_3_), 0.78 (s, 9H, SiC(CH_3_)_3_), −0.08 (s, 3H, Si(CH_3_)), −0.29 (s, 3H, Si(CH_3_)); ^13^C-NMR (CDCl_3_, 100 MHz) δ: 191.0, 170.9, 167.5, 166.2, 148.9, 142.1, 137.9, 133.9, 133.1, 131.9, 131.4 (q, ^2^*J*_C-F_ = 33.3 Hz), 130.2, 128.7, 128.0, 126.9, 126.7 (q, ^3^*J*_C-F_ = 4.1 Hz), 126.5, 123.5 (q, ^1^*J*_C-F_ = 273.7 Hz), 87.2, 86.3, 80.6, 75.8, 75.6, 74.5, 70.4, 67.6, 59.0, 55.7, 44.4, 43.2, 38.3, 36.6, 29.7, 29.1, 25.5, 22.1, 21.5, 18.2, 14.8, 14.4, −5.5, −6.0; LC-MS (ESI, *m/z*): [M + 1], found 994.3, [M + 23], found 1016.3, C_52_H_62_F_3_NO_13_Si: 993.39. 

#### 3.1.4. General Experimental Procedure for Compounds **16a**–**e**

*10-Dehydro-7,8-seco-10-deacetylpaclitaxel* (**16a**). To a solution of of **15a** (19.8 mg, 0.0214 mmol) in THF (1.4 mL) was added, dropwise, 0.3 mL of HF-pyridine (*v/v* = 1:2) at 0 °C, and the reaction mixture was stirred at room temperature for 30 h. The reaction was quenched with saturated aqueous NaHCO_3_, diluted with ethyl acetate, washed with saturated NH_4_Cl. The organic layer dried over anhydrous Na_2_SO_4_ and then concentrated under reduced pressure. Purification of the crude product by silica gel chromatography (ethyl acetate: petroleum ether = 2:1) gave product **16a** as a white solid (13.4 mg, 77.4% yield). ^1^H-NMR (CDCl_3_, 400 MHz) δ: 8.04 (d, *J* = 7.6 Hz, 2H, Ar-H), 7.69 (d, *J* = 7.2 Hz, 2H, Ar-H), 7.57 (t, *J* = 7.4 Hz, 1H, Ar-H), 7.48–7.44 (m, 5H, Ar-H), 7.38–7.27 (m, 6H, NH, Ar-H), 6.47 (s, 1H, OH-9), 6.16 (t, *J* = 7.5 Hz, 1H, H-13), 5.86 (dd, *J* = 2.8, 9.4 Hz, 1H, H-3′), 5.60 (d, *J* = 9.5 Hz, 1H, H-2), 5.20 (brs, 1H, H-5), 5.16 (d, *J* = 11.4 Hz, 1H, H-20a), 4.78 (d, *J* = 3.0 Hz, 1H, H-2′), 4.27 (d, *J* = 8.1 Hz, 1H, H-3), 4.20 (brs, 1H, H-20b), 3.89 (brs, 1H, H-7a), 3.68 (dt, *J* = 6.2, 10.1 Hz, 1H, H-7b), 2.77 (brs, 1H, H-6a), 2.61 (brs, 1H, H-14a), 2.38 (dd, *J* = 9.7, 15.9 Hz, 1H, H-14b), 2.29 (brs, 1H, OH), 2.10–2.07 (m, 1H, H-6b), 1.83 (brs, 6H, 2 × CH_3_), 1.75 (s, 3H, CH_3_), 1.20 (s, 3H, CH_3_), 1.07 (s, 3H, CH_3_); ^13^C-NMR (CDCl_3_, 100 MHz) δ: 191.0, 172.4, 169.0, 167.3, 167.2, 148.8, 142.2, 138.0, 136.7, 133.8, 133.6, 132.0, 129.7, 129.3, 129.0, 128.8, 128.7, 128.1, 127.1, 127.0, 124.6, 86.9, 86.2, 80.5, 74.8, 73.7, 70.7, 59.6, 54.8, 43.0, 36.6, 29.7, 24.9, 22.1, 21.2, 18.4, 14.6, 14.5; LC-MS (ESI, *m/z*): [M + 1], found 812.0, [M + 23], found 834.0, C_45_H_49_NO_13_: 811.32. 

*2-Debenzoyl-2-(3-methoxybenzoyl)-10-dehydro-7,8-seco-10-deacetylpaclitaxel* (**16b**). White solid; yield 65.2%; ^1^H-NMR (CDCl_3_, 400 MHz) δ: 7.72 (d, *J* = 7.6 Hz, 2H, Ar-H), 7.60 (d, *J* = 7.1 Hz, 1H, Ar-H), 7.47–7.43 (m, 5H, Ar-H), 7.36–7.31 (m, 5H, Ar-H), 7.29–7.25 (m, 1H, Ar-H), 7.10 (d, *J* = 8.1 Hz, 1H, NH), 6.49 (brs, 1H, OH-9), 6.09 (t, *J* = 7.4 Hz, 1H, H-13), 5.81 (d, *J* = 9.2 Hz, 1H, H-3′), 5.58 (d, *J* = 9.5 Hz, 1H, H-2), 5.20 (brs, 1H, H-5), 5.14 (d, *J* = 11.3 Hz, 1H, H-20a), 4.73 (brs, 1H, H-2′), 4.24 (d, *J* = 7.2 Hz, 1H, H-3), 4.19 (brs, 1H, H-20b), 3.87 (brs, 1H, H-7a), 3.79 (s, 3H, OCH_3_), 3.65 (brs, 1H, H-7b), 2.74 (brs, 1H, H-6a), 2.56 (brs, 1H, H-14a), 2.40 (dd, *J* = 9.9, 15.9 Hz, 1H, H-14b), 2.35 (brs, 1H, OH-1), 2.09 (m, 1H, H-6b), 1.83 (brs, 6H, 2 × CH_3_), 1.69 (s, 3H, CH_3_), 1.17 (s, 3H, CH_3_), 1.06 (s, 3H, CH_3_); ^13^C-NMR (CDCl_3_, 100 MHz) δ: 191.1, 172.4, 169.0, 167.1, 159.8, 148.8, 142.1, 138.1, 136.7, 133.6, 131.9, 130.7, 130.1, 129.0, 128.7, 128.6, 128.2, 128.1, 127.1, 124.5, 121.8, 119.5, 114.8, 86.9, 86.1, 80.4, 74.8, 73.7, 70.5, 59.7, 55.5, 55.0, 44.5, 42.9, 36.6, 29.8, 29.7, 24.9, 22.2, 21.1, 14.6, 14.5; LC-MS (ESI, *m/z*): [M + 1], found 842.4, [M + 23], found 864.3, C_46_H_51_NO_14_: 841.33. 

*2-Debenzoyl-2-(3-fluorobenzoyl)-10-dehydro-7, 8-seco-10-deacetylpaclitaxel* (**16c**). White solid; yield 59.3%, ^1^H-NMR (CDCl_3_, 300 MHz) δ: 7.90 (d, *J* = 7.7 Hz, 1H, Ar-H), 7.75–7.68 (m, 3H, Ar-H), 7.48–7.44 (m, 4H, Ar-H), 7.41–7.31 (m, 6H, Ar-H), 7.24 (d, *J* = 8.0 Hz, 1H, NH), 6.46 (brs, 1H, OH-9), 6.20 (t, *J* = 7.9Hz, 1H, H-13), 5.88 (d, *J* = 7.6 Hz, 1H, H-3′), 5.59 (d, *J* =9.5 Hz, 1H, H-2), 5.19 (overlap, *J* = 11.5 Hz, 2H, H-20a, H-5), 4.81 (brs, 1H, H-2′), 4.46 (brs, 1H, OH), 4.29 (d, *J* = 7.9Hz, 1H, H-3), 4.22 (brs, 1H, H-20b), 3.92 (brs, 1H, H-7a), 3.70 (brs, 1H, H-7b), 3.39 (brs, 1H, OH), 2.77 (brs, 1H, H-6a), 2.60 (brs, 1H, H-14a), 2.37 (dd, *J* = 9.7, 15.8 Hz, 1H, H-14b), 2.09 (m, 1H, H-6b), 1.85 (brs, 6H, 2 × CH_3_), 1.78 (s, 3H, CH_3_), 1.64 (brs, 1H, OH-1), 1.22 (s, 3H, CH_3_), 1.09 (s, 3H, CH_3_); ^13^C-NMR (CDCl_3_, 75 MHz) δ: 190.9, 172.4, 167.2, 166.2, 162.7 (d, ^1^*J*_C-F_ = 248.7 Hz), 148.9, 142.1, 137.9, 133.6, 132.0, 131.4 (d, ^2^*J*_C-F_ = 7.4 Hz), 130.8 (d, ^3^*J*_C-F_ = 7.9 Hz), 128.9, 128.7, 128.2, 127.0, 126.9, 125.5, 124.4, 121.0 (d, ^2^*J*_C-F_ = 21.2 Hz), 116.5 (d, ^2^*J*_C-F_ = 23.5 Hz), 86.8, 86.2, 80.5, 75.2, 74.7, 73.6, 70.7, 59.5, 54.6, 42.9, 36.5, 29.7, 29.3, 24.9, 22.1, 21.2, 14.7, 14.5; LC-MS (ESI, *m/z*): [M + 1], found 830.3, [M + 23], found 852.3, C_45_H_48_FNO_13_: 829.31. 

*2-Debenzoyl-2-(3-chlorobenzoyl)-10-dehydro-7, 8-seco-10-deacetylpaclitaxel* (**16d**). White solid; yield 60.3%; ^1^H-NMR (CDCl_3_, 400 MHz) δ: 7.98 (d, *J* = 7.5 Hz, 2H, Ar-H), 7.72 (d, *J* = 7.0 Hz, 2H, Ar-H), 7.56 (d, *J* = 8.4 Hz, 1H, Ar-H), 7.47–7.42 (m, 4H, Ar-H), 7.40–7.29 (m, 6H, NH, Ar-H), 6.49 (brs, 1H, OH-9), 6.15 (t, *J* = 7.8 Hz, 1H, H-13), 5.86 (d, *J* = 8.1 Hz, 1H, H-3′), 5.58 (d, *J* = 9.5 Hz, 1H, H-2), 5.21 (brs, 1H, H-5), 5.14 (d, *J* =11.0 Hz, 1H, H-20a), 4.79 (d, *J* = 2.8 Hz, 1H, H-2′), 4.27 (d, *J* =7.8 Hz, 1H, H-3), 4.20 (brs, 1H, H-20b), 3.89 (brs, 1H, H-7a), 3.71–3.67 (m, 1H, H-7b), 2.75 (brs, 1H, H-6a), 2.59 (brs, 1H, H-14a), 2.38 (dd, *J* = 9.7, 15.8 Hz, 1H, H-14b), 2.25 (brs, 1H, OH), 2.09 (m, 1H, H-6b), 2.00 (m, 1H, OH), 1.84 (brs, 6H, 2 × CH_3_), 1.74 (s, 3H, CH_3_), 1.20 (s, 3H, CH_3_), 1.07 (s, 3H, CH_3_); ^13^C-NMR (CDCl_3_, 100 MHz) δ: 191.0, 172.5, 167.2, 166.0, 148.9, 142.1, 138.1, 135.0, 133.8, 133.7, 132.0, 131.1, 130.4, 129.5, 128.8, 128.7, 128.1, 127.9, 127.0, 124.3, 86.9, 86.2, 80.4, 75.3, 74.7, 73.6, 70.6, 59.6, 54.8, 50.8, 42.9, 36.6, 29.7, 29.5, 29.3, 24.9, 22.2, 21.2, 14.6, 14.5; LC-MS (ESI, *m/z*): [M + 1], found 845.9, [M + 23], found 867.8, C_45_H_48_ClNO_13_: 845.28. 

*2-Debenzoyl-2-(3-trifluoromethylbenzoyl)-10-dehydro-7,8-seco-10-deacetylpaclitaxel* (**16e**). White solid; yield 55.5%; ^1^H-NMR (CDCl_3_, 400 MHz) δ: 8.33 (d, *J* = 7.7 Hz, 1H, Ar-H), 8.25 (brs, 1H, NH), 7.85 (d, *J* = 7.4 Hz, 1H, Ar-H), 7.68–7.62 (m, 3H, Ar-H), 7.48–7.45 (m, 3H, Ar-H), 7.40–7.29 (m, 6H, Ar-H), 6.47 (brs, 1H, OH-9), 6.18 (t, *J* = 7.9 Hz, 1H, H-13), 5.87 (d, *J* = 7.6 Hz, 1H, H-3′), 5.62 (d, *J* = 9.4 Hz, 1H, H-2), 5.18 (overlap, *J* = 11.5 Hz, 2H, H-20a, H-5), 4.80 (d, *J* = 2.8 Hz, 1H, H-2′), 4.62 (brs, 1H, OH), 4.28 (d, *J* = 7.6 Hz, 1H, H-3), 4.24 (brs, 1H, H-20b), 3.90 (brs, 1H, H-7a), 3.68 (brs, 1H, H-7b), 3.38 (brs, 1H, OH), 2.77 (brs, 1H, H-6a), 2.58 (brs, 1H, H-14a), 2.37 (dd, *J* = 9.7, 15.7 Hz, 1H, H-14b), 2.05 (brs, 1H, OH), 2.03–1.98 (m, 1H, H-6b), 1.84 (brs, 6H, 2 × CH_3_), 1.78 (s, 3H, CH_3_), 1.22 (s, 3H, CH_3_), 1.09 (s, 3H, CH_3_); ^13^C-NMR (CDCl_3_, 100 MHz) δ: 190.9, 172.4, 167.2, 166.0, 148.9, 142.1, 138.0, 133.7, 133.0, 132.0, 131.7, 130.3, 130.0, 129.7, 128.9, 128.7, 128.2, 127.0 (2C), 126.4, 124.8, 124.2, 122.1, 86.9, 86.2, 80.5, 75.4, 74.7, 73.5, 70.7, 59.6, 54.7, 43.0, 36.6, 31.9, 29.7, 25.5, 22.1, 21.2, 14.6, 14.4; LC-MS (ESI, *m/z*): [M + 1], found 880.0, [M + 23], found 901.9, C_46_H_48_F_3_NO_13_: 879.31. 

#### 3.1.5. General Experimental Procedure for Compounds **17a**–**e**

*2′-O-(tert-Butyldimethylsilyl)-9-O-(2-hydroxyethyl)-10-dehydro-7, 8-seco-10-deacetylpaclitaxel* (**17a**). To a mixture of **15a** (50 mg, 0.0540 mmol), potassium carbonate (104 mg, 0.754 mmol), and potassium iodide (18 mg, 0.108 mmol) in DMF (0.9 mL) was added 2-bromoethanol (54 μL 0.762 mmol), dropwise, at 0 °C. The reaction mixture was stirred for 9 h at room temperature and then was diluted with ethyl acetate. The organic phase was washed with saturated aqueous NH_4_Cl. The organic layer was dried over anhydrous Na_2_SO_4_ and then concentrated under reduced pressure. Purification of the crude product by silica gel chromatography (ethyl acetate: petroleum ether = 1:1) gave product **17a** as a white solid (19 mg, 52.4% yield). ^1^H-NMR (CDCl_3_, 300 MHz) δ: 8.09 (d, *J* = 7.2 Hz, 2H, Ar-H), 7.62 (d, *J* = 7.2 Hz, 2H, Ar-H), 7.57 (d, *J* = 7.5 Hz, 1H, Ar-H), 7.49 (t, *J* = 7.5 Hz, 2H, Ar-H), 7.36 (m, 8H, Ar-H), 6.30 (t, *J* = 8.4 Hz, 1H, H-13), 5.88 (d, *J* = 9.0 Hz, 1H, H-3′), 5.57 (d, *J* = 9.6 Hz, 1H, H-2), 5.28 (overlap, 2H, H-5, H-20a), 4.69 (d, *J* = 1.8 Hz, 1H, H-2′), 4.26 (d, *J* = 7.8 Hz, 1H, H-3), 4.20 (m, 2H, H-20b, OCH_2_CH_2_OCO), 3.94 (m, 1H, OCH_2_CH_2_OCO), 3.87 (m, *J* = 1.8, 6.3, 8.7, 12.0 Hz, 1H, H-7a), 3.74 (overlap, 2H, H-7b, OCH_2_CH_2_OCO), 3.62 (m, 1H, OCH_2_CH_2_OCO), 2.71 (m, 2H, H-6a, H-14a), 2.25 (dd, *J* = 9.3, 15.9 Hz, 2H, H-6b, H-14b), 1.97 (s, 3H, CH_3_), 1.92 (s, 3H, CH_3_), 1.82 (s, 3H, CH_3_), 1.27 (s, 3H, CH_3_), 1.16 (s, 3H, CH_3_), 0.76 (s, 9H, SiC(CH_3_)_3_), −0.08 (s, 3H, Si(CH_3_)), −0.33 (s, 3H, Si(CH_3_)); ^13^C-NMR (CDCl_3_, 100 MHz) δ: 192.3, 170.7, 168.7, 167.3, 153.8, 144.0, 139.4, 138.0, 133.7 (2C), 131.8, 129.9 (2C), 128.9 (2C), 128.6 (4C), 127.9, 126.9 (2C), 126.4 (2C), 87.1, 87.7, 85.7, 79.9, 75.6, 75.4, 74.6, 73.8, 70.5, 61.8, 58.7, 55.6, 42.9, 36.6, 25.4 (3C), 22.9, 22.0, 18.1, 15.3, 14.6, −5.5, −6.1; LC-MS (ESI, *m/z*): [M + 1], found 970.5, [M + 23], found 992.4, C_53_H_67_NO_14_Si: 969.43. 

*2′-O-(tert-Butyldimethylsilyl)-2-debenzoyl-2-(3-methoxybenzoyl)-9-O-(2-hydroxyethyl)-10-dehydro-7,8-seco-10-deacetylpaclitaxel* (**17b**). White solid; yield 56.1%; ^1^H-NMR (CDCl_3_, 400 MHz) δ: 7.72–7.66 (m, 3H, Ar-H), 7.54 (s, 1H, Ar-H), 7.47–7.43 (m, 2H, Ar-H), 7.42–7.29 (m, 8H, Ar-H), 7.12 (dd, *J* = 2.0, 8.4 Hz, 1H, BzNH), 6.29 (t, *J* = 8.5 Hz, 1H, H-13), 5.88 (d, *J* = 9.3 Hz, 1H, H-3′), 5.59 (d, *J* = 9.6 Hz, 1H, H-2), 5.26 (overlap, 2H, H-5, H-20a), 4.72 (s, 1H, H-2′), 4.26 (d, *J* = 6.4 Hz, 1H, H-3), 4.19–4.15 (m, 1H, H-20b), 3.94–3.64 (m, 9H, OCH_3_, OCH_2_CH_2_OH, H-7a, H-7b), 2.61 (br, 2H, H-6b, H-14b), 2.25 (dd, *J* = 9.6, 15.6 Hz, 2H, H-6a, H-14a), 2.03 (s, 3H, OAc), 1.93 (s, 6H, 2 × CH_3_), 1.26 (s, 3H, CH_3_), 1.16 (s, 3H, CH_3_), 0.77 (s, 9H, SiC(CH_3_)_3_), −0.10 (s, 3H, Si(CH_3_)), −0.30 (s, 3H, Si(CH_3_)); ^13^C-NMR (CDCl_3_, 100 MHz) δ: 192.4, 171.2, 170.9, 168.8, 167.3, 167.0, 162.6, 159.7, 144.1, 138.1, 133.8, 131.8, 130.4, 130.2, 128.6 (2C), 127.9, 127.0 (2C), 126.6 (2C), 122.2, 119.5, 114.9, 85.7, 79.8, 75.8, 74.8, 61.8, 58.9, 55.7, 55.5, 42.9, 36.6, 36.5, 31.9, 31.4, 29.7, 25.5 (3C), 22.9, 22.1, 18.2, 15.2, 14.6, −5.5, −6.1; LC-MS (ESI, *m/z*): [M + 1], found 1000.5, [M + 23], found 1022.5, C_54_H_69_NO_16_Si: 999.44. 

*2′-O-(tert-Butyldimethylsilyl)-2-debenzoyl-2-(3-fluorobenzoyl)-9-O-(2-hydroxyethyl)-10-dehydro-7,8-seco-10-deacetylpaclitaxel* (**17c**). White solid; yield 53.2%; ^1^H-NMR (CDCl_3_, 400 MHz) δ: 7.93 (d, *J* = 7.8 Hz, 1H, Ar-H), 7.76 (d, *J* = 8.8 Hz, 1H, Ar-H), 7.63 (d, *J* = 7.6 Hz, 2H, Ar-H), 7.52–7.28 (m, 11H, NH, Ar-H), 6.29 (t, *J* = 8.1 Hz, 1H, H-13), 5.89 (d, *J* = 9.2 Hz, 1H, H-3′), 5.58 (d, *J* = 9.6 Hz, 1H, H-2), 5.35–5.15 (overlap, 2H, H-5, H-20a), 4.70 (d, *J* = 1.9 Hz, 1H, H-2′), 4.26 (d, *J* = 7.6 Hz, 1H, H-3), 4.20–4.16 (m, 2H, H-20b, OCH_2_CH_2_OCO), 3.95 (brs, 1H, OCH_2_CH_2_OCO), 3.89–3.84 (ddd, *J* = 2.3, 6.8, 8.7, 12.4 Hz, 1H, H-7a), 3.80–3.70 (overlap, 2H, H-7b, OCH_2_CH_2_OCO), 3.60–3.58 (m, 1H, OCH_2_CH_2_OCO), 2.71–2.63 (m, 2H, H-6a, H-14a), 2.25 (dd, *J* = 9.6, 15.8 Hz, 2H, H-6b, H-14b), 2.03 (s, 3H, CH_3_), 1.91(s, 3H, CH_3_), 1.87 (brs, 3H, CH_3_), 1.83 (brs, 2H, 2 × OH), 1.26 (s, 3H, CH_3_), 1.15 (s, 3H, CH_3_), 0.73 (s, 9H, SiC(CH_3_)_3_), −0.08 (s, 3H, Si(CH_3_)), −0.32 (s, 3H, Si(CH_3_)); ^13^C-NMR (CDCl_3_, 100 MHz) δ: 192.3, 171.2, 170.8, 168.6, 167.4, 166.2, 162.7 (d, ^1^*J*_C-F_ = 248.3 Hz), 144.0, 137.9, 133.9, 131.9, 131.2 (d, ^3^*J*_C-F_ = 7.4 Hz), 131.0 (d, ^3^*J*_C-F_ = 7.7 Hz), 128.7 (2C), 128.0, 127.0, 126.5, 125.8 (d, ^4^*J*_C-F_ = 2.6 Hz), 121.0 (d, ^2^*J*_C-F_ = 21.2 Hz), 116.3 (d, ^2^*J*_C-F_ = 22.9 Hz), 85.8, 80.0, 75.7, 75.2, 70.5, 61.9, 58.7, 55.6, 45.7, 42.9, 36.7, 25.5, 22.9, 22.1, 18.2, 15.3, 14.6, 13.7, −5.4, −6.0; LC-MS (ESI, *m/z*): [M + 1], found 988.0, C_53_H_66_FNO_14_Si: 987.42.

*2′-O-(tert-Butyldimethylsilyl)-2-debenzoyl-2-(3-chlorobenzoyl)-9-O-(2-hydroxyethyl)-10-dehydro-7,8-seco-10-deacetylpaclitaxel* (**17d**). White solid; yield 60.9%; ^1^H-NMR (CDCl_3_, 400 MHz) δ: 8.06 (d, *J* = 8.8 Hz, 2H, Ar-H), 7.63 (d, *J* = 7.5 Hz, 2H, Ar-H), 7.57 (d, *J* = 8.2 Hz, 1H, Ar-H), 7.50–7.43 (m, 3H, Ar-H), 7.43–7.33 (m, 8H, Ar-H, BzNH), 6.31 (t, *J* = 8.0 Hz, 1H, H-13), 5.90 (d, *J* = 9.2 Hz, 1H, H-3′), 5.59 (d, *J* = 9.6 Hz, 1H, H-2), 5.32–5.28 (overlap, 2H, H-5, H-20b), 4.71 (d, *J* = 1.9 Hz, 1H, H-2′), 4.28 (d, *J* = 7.6 Hz, 1H, H-3), 4.24–4.20 (m, 2H, H-20a, OCH_2_CH_2_OH), 3.98–3.95 (m, 1H, OCH_2_CH_2_OH), 3.93–3.89 (m, 1H, H-7b), 3.78–3.72 (overlap, 2H, H-7a, OCH_2_CH_2_OH), 3.65 (br, 1H, OCH_2_CH_2_OH), 2.72–2.68 (m, 2H, H-6b, H-14b), 2.26 (dd, *J* = 7.2, 13.4 Hz, 2H, H-6a, H-14a), 1.98 (s, 3H, CH_3_), 1.93 (brs, 1H, OH), 1.85 (brs, 2H, OH), 1.29 (brs, 6H, CH_3_), 1.18 (s, 3H, CH_3_), 0.78 (s, 9H, SiC(CH_3_)_3_), −0.08 (s, 3H, Si(CH_3_)), −0.30 (s, 3H, Si(CH_3_)); ^13^C-NMR (CDCl_3_, 100 MHz) δ: 192.3, 170.8, 167.4, 166.3, 144.0, 137.9, 135.0, 133.9 (2C), 132.0, 130.8, 130.7, 129.7, 128.8 (2C), 128.7 (2C), 128.2, 128.1, 127.0 (2C), 126.6 (2C), 85.9, 75.8, 75.3, 70.5, 62.0, 55.6, 43.0, 36.7, 32.0, 29.7, 29.4, 27.3, 25.5 (3C), 23.0, 22.7, 22.2, 18.3, 15.3, 14.7, 14.2, −5.4, −6.0; LC-MS (ESI, *m/z*): [M + 1], found 1004.4, [M + 23], found 1026.4, C_53_H_66_ClNO_14_Si: 1003.39.

*2′-O-(tert-Butyldimethylsilyl)-2-debenzoyl-2-(3-trifluoromethylbenzoyl)-9-O-(2-hydroxyethyl)-10-dehydro-7, 8-seco-10-deacetylpaclitaxel* (**17e**). White solid; yield 49.9%; ^1^H-NMR (CDCl_3_, 400 MHz) δ: 7.83 (d, *J* = 7.7 Hz, 2H, Ar-H), 7.68 (t, *J* = 7.8 Hz, 1H, Ar-H), 7.58 (d, *J* = 7.3 Hz, 1H, Ar-H), 7.45 (t, *J* = 7.4 Hz, 2H, Ar-H), 7.35 (m, 8H, Ar-H), 6.29 (t, *J* = 8.2 Hz, 1H, H-13), 5.87 (d, *J* = 8.7 Hz, 1H, H-3′), 5.60 (d, *J* = 9.65 Hz, 1H, H-2), 5.26 (overlap, 2H, H-5, H-20a), 4.70 (d, *J* = 1.3 Hz, 1H, H-2′), 4.28 (d, *J* = 6.8 Hz, 1H, H-3), 4.19 (m, 2H, H-20b, OCH_2_CH_2_OCO), 3.98 (m, 1H, OCH_2_CH_2_OCO), 3.87 (m, *J* = 1.8, 6.3, 8.7, 12.0 Hz, 1H, H-7a), 3.75 (overlap, 2H, H-7b, OCH_2_CH_2_OCO), 3.66 (m, 1H, OCH_2_CH_2_OCO), 2.65 (m, 2H, H-6a, H-14a), 2.24 (dd, *J* = 6.7, 16.3 Hz, 2H, H-6b, H-14b), 1.97 (s, 3H, CH_3_), 1.92 (s, 3H, CH_3_), 1.82 (s, 3H, CH_3_), 1.26 (s, 3H, CH_3_), 1.17 (s, 3H, CH_3_), 0.77 (s, 9H, SiC(CH_3_)_3_), −0.08 (s, 3H, Si(CH_3_)), −0.32 (s, 3H, Si(CH_3_)); ^13^C-NMR (CDCl_3_, 100 MHz) δ: 192.3, 170.8, 167.6, 166.1, 144.1, 137.8, 133.9, 133.2, 132.0, 131.4 (q, ^2^J_C-F_ = 33.2 Hz), 130.3, 130.2 (q, ^3^J_C-F_ = 3.5 Hz), 130.0, 128.8, 128.7, 128.1, 126.9, 126.8 (q, ^3^J_C-F_ = 3.6 Hz), 126.5, 123.6 (q, ^1^J_C-F_ = 273.8 Hz), 85.8, 75.6, 75.4, 70.5, 62.0, 58.4, 55.6, 43.0, 36.7, 31.9, 29.8, 29.7, 29.6 (2C), 29.3 (2C), 27.2, 25.5, 23.0, 22.7, 22.1, 18.4, 18.2, 15.3, 14.6, 14.1, −5.4, −6.0; LC-MS (ESI, *m/z*): [M + 1], found 1038.4, [M + 23], found 1060.4, C_54_H_66_F_3_NO_14_Si: 1037.42.

#### 3.1.6. General Experimental Procedure for Compounds **19a**–**e**

*2′-O-(tert-Butyldimethylsilyl)-macrocyclic taxoid* (**19a**). To a solution of **17a** (19 mg, 0.0196 mmol) in dichloromethane (4.2 mL) was added, dropwise, pyridine (22 μL, 0.266 mmol) and triphosgene (8.7 mg, 0.0293 mmol) at 0 °C, then the reaction mixture was stirred at room temperature for 5 h. The reaction was diluted with ethyl acetate, washed with saturated NaCl. The organic layer was dried over anhydrous Na_2_SO_4_ and then concentrated under reduced pressure. Purification of the crude product by silica gel chromatography (ethyl acetate: petroleum ether = 1:3) gave product **19a** as a white solid (3.1 mg, 15.9% yield). ^1^H-NMR (CDCl_3_, 400 MHz) δ: 8.17 (d, *J* = 7.2 Hz, 2H, Ar-H), 7.75 (d, *J* = 7.2 Hz, 2H, Ar-H), 7.63 (t, *J* = 7.2 Hz, 1H, Ar-H), 7.54 (t, *J* = 7.6 Hz, 2H, Ar-H), 7.50 (d, *J* = 7.6 Hz, 1H, Ar-H), 7.43 (d, *J* = 7.6 Hz, 2H, Ar-H), 7.39 (t, *J* = 7.2 Hz, 2H, Ar-H), 7.34–7.29 (m, 3H, Ar-H), 7.08 (d, *J* = 9.2 Hz, 1H, NH), 6.27 (t, *J* = 8.8 Hz, 1H, H-13), 5.76 (d, *J* = 8.4 Hz, 1H, H-3′), 5.60 (d, *J* = 7.6 Hz, 1H, H-2), 4.84 (overlap, 2H, H-5, OCH_2_CH_2_OCO), 4.79 (d, *J* = 7.6 Hz, 1H, H-3), 4.67 (d, *J* = 2.0 Hz, 1H, H-2′), 4.57 (dd, *J* = 6.8, 12.0 Hz, 1H, OCH_2_CH_2_OCO), 4.49 (dd, *J* = 6.8, 12.0 Hz, 1H, OCH_2_CH_2_OCO), 4.46 (d, *J* = 8.4 Hz, 1H, H-20a), 4.35 (d, *J* = 8.4 Hz, 1H, H-20b), 4.32 (dd, *J* = 5.2, 13.6 Hz, 1H, OCH_2_CH_2_OCO), 4.13 (ddd, *J* = 6.4, 10.8, 17.2 Hz, 1H, H-7a), 4.02 (ddd, *J* = 8.8, 11.6, 17.2 Hz, 1H, H-7b), 2.55 (s, 3H, 4-OAc), 2.35 (m, 3H, H-6a, H-6b, H-14a), 2.18 (dd, *J* = 8.8, 15.2 Hz, 1H, H-14b), 1.84 (s, 3H, CH_3_), 1.25 (s, 3H, CH_3_), 1.24 (s, 3H, CH_3_), 1.19 (s, 3H, CH_3_), 0.79 (s, 9H, SiC(CH_3_)_3_), −0.03 (s, 3H, Si(CH_3_)), −0.30 (s, 3H, Si(CH_3_)); ^13^C-NMR (CDCl_3_, 150 MHz) δ: 192.1, 171.0, 169.4, 166.9, 166.8, 152.4, 151.5, 144.5, 139.7, 138.3, 134.6, 134.1, 133.8, 131.8, 130.2 (2C), 129.0, 128.8 (2C), 128.7, 127.9, 127.0 (2C), 126.3 (2C), 89.6, 84.7, 78.6, 76.1, 75.3, 75.2, 71.4, 70.6, 68.9, 65.9, 55.6, 42.9, 42.4, 36.5, 31.0, 26.2, 25.5 (3C), 22.9, 21.6, 18.1, 14.4, 13.6, −5.3, −5.9; LC-MS (ESI, *m/z*): [M + 1], found 996.4, [M + 23], found 1018.4, C_54_H_65_NO_15_Si: 995.41.

*2′-O-(tert-Butyldimethylsilyl)-2-debenzoyl-2-(3-methoxybenzoyl)-macrocyclic taxoid* (**19b**). White solid; yield 32.7%; ^1^H-NMR (CDCl_3_, 400 MHz) δ: 7.80 (d, *J* = 1.2, 7.6 Hz, 1H, Ar-H), 7.77 (d, *J* = 6.8 Hz, 2H, Ar-H), 7.67 (dd, *J* = 1.5, 2.5 Hz, 1H, Ar-H), 7.55–7.51 (m, 1H, Ar-H), 7.49–7.39 (m, 3H, Ar-H), 7.40–7.31 (m, 5H, Ar-H), 7.20–7.17 (m, 1H, Ar-H), 7.08 (d, *J* = 9.2 Hz, 1H, BzNH), 6.28 (t, 1H, H-13), 5.76 (dd, *J* = 1.6, 8.9 Hz, H-3′), 5.62 (d, *J* = 7.5 Hz, 1H, H-2), 4.87 (d, *J* = 7.6 Hz, 1H, H-20b), 4.85 (d, *J* = 4.8 Hz, 1H, H-5), 4.81 (d, *J* = 7.5 Hz, 1H, H-20a), 4.68 (d, *J* = 2.0 Hz, 1H, H-2′), 4.60–4.32 (m, 5H, H-7a, H-7b, H-16a, H-16b, H-3), 4.18–4.03 (m, H-15a, H-15b), 3.91 (s, 3H, OCH_3_), 2.55 (d, *J* = 1.3 Hz, 6H, 2 × CH_3_), 2.41–2.35 (m, 3H, H-14b, H-6a, H-6b), 2.24–2.20 (m, 1H, H-14a), 1.84 (d, *J* = 1.2 Hz, 3H, COCH_3_), 1.26 (s, 3H, CH_3_), 1.21 (s, 3H, CH_3_), 0.81 (s, 9H, SiC(CH_3_)_3_), −0.008 (s, 3H, Si(CH_3_)), −0.28 (s, 3H, Si(CH_3_)); ^13^C-NMR (CDCl_3_, 100 MHz) δ: 192.2, 171.1, 169.4, 166.9, 166.8, 159.8, 152.5, 151.6, 144.6, 139.7, 138.4, 134.7, 134.2, 131.9, 130.4, 130.0, 128.82 (2C), 128.78 (2C), 128.0, 127.0 (2C), 126.4 (2C), 122.6, 120.3, 114.7, 89.6, 84.8, 78.5, 75.4, 75.3, 71.5, 70.7, 69.0, 65.9, 55.6, 43.0, 42.5, 36.5, 31.1, 29.7, 26.3, 25.6 (3C), 22.9, 21.6, 18.2, 14.4, 13.7, −5.2,−5.8; LC-MS (ESI, *m/z*): [M + 1], found 1026.4, [M + 23], found 1048.4, C_55_H_67_NO_16_Si: 1025.42.

*2′-O-(tert-Butyldimethylsilyl)-2-debenzoyl-2-(3-fluorobenzoyl)-macrocyclic taxoid* (**19c**). White solid; yield 32.1%; ^1^H-NMR (CDCl_3_, 400 MHz) δ: 8.01(d, *J* = 7.8 Hz, 1H, Ar-H), 7.84–7.87 (m, 1H, Ar-H), 7.75–7.73 (m, 2 H, Ar-H), 7. 56–7.49 (m, 2H, Ar-H), 7.44–7.31 (m, 8H, Ar-H), 7.09 (d, *J* = 9.0 Hz, 1H, NH), 6.28 (t, *J* = 8.8 Hz,1H, H-13), 5.78 (dd, *J* = 1.4, 9.1 Hz, 1H, H-3′), 5.61 (d, *J* = 7.6 Hz, 1H, H-2), 4.87–4.82 (overlap, 2H, OCH_2_CH_2_OCO, H-5), 4.83 (d, *J* = 7.5 Hz, 1H, H-3), 4.69 (d, *J* = 2.0 Hz, 1H, H-2′), 4.58 (dd, *J* = 6.4, 11.9 Hz, 1H, OCH_2_CH_2_OCO), 4.51 (dd, *J* = 6.4, 13.7 Hz, 1H, OCH_2_CH_2_OCO), 4.46 (d, *J* = 8.2 Hz, 1H, H-20a), 4.36–4.30 (overlap, 2H, OCH_2_CH_2_OCO, H-20b), 4.16–4.09 (m, 1H, H-7a), 4.06–3.99 (m, 17.2 Hz, 1H, H-7b), 2.57 (s, 3H, COCH_3_), 2.56 (s, 3H, 4-OAc), 2.39–2.31 (m, 3H, H-6a, H-6b, H-14a), 2.18 (dd, *J* = 8.4, 15.4 Hz, 1H, H-14b), 1.90 (brs, 1H, OH), 1.84 (s, 3H, CH_3_), 1.23 (s, 3H, CH_3_), 1.19 (s, 3H, CH_3_), 0.80 (s, 9H, SiC(CH_3_)_3_), −0.29 (s, 3H, Si(CH_3_)), −0.03 (s, 3H, Si(CH_3_)); ^13^C-NMR (CDCl_3_, 100 MHz) δ: 192.1, 171.1, 169.4, 167.0, 165.7, 162.7 (d, ^1^*J*_C-F_ = 247.8 Hz), 152.5, 151.6, 144.5, 139.8, 138.4, 134.5, 134.2, 131.9, 131.3 (d, ^3^*J*_C-F_ = 7.6Hz), 130.6 (d, ^3^*J*_C-F_ = 7.6 Hz), 128.8 (2C), 128.0, 127.0, 126.5, 126.0 (d, ^4^*J*_C-F_ = 3.0 Hz), 121.0 (d, ^2^*J*_C-F_ = 21.4 Hz), 117.1 (d, ^2^*J*_C-F_ = 23.3 Hz), 89.6, 84.7, 78.6, 76.1, 75.7, 75.3, 71.5, 70.7, 68.9, 65.9, 55.6, 43.0, 42.5, 36.4, 31.0, 26.3, 25.6, 22.8, 21.6, 18.2, 14.4, 13.6, −5.2, −5.8; LC-MS (ESI, *m/z*): [M + 1], found 1014.4, [M + 23], found 1036.4, C_54_H_64_FNO_15_Si: 1013.40.

*2′-O-(tert-Butyldimethylsilyl)-2-debenzoyl-2-(3-chlorobenzoyl)-macrocyclic taxoid* (**19d**). White solid; yield 36.1%; ^1^H-NMR (CDCl_3_, 400 MHz) δ: 8.19 (t, *J* = 1.6 Hz, 1H, Ar-H), 8.10–8.07 (m, 1H, Ar-H), 7.77–7.75 (m, 2H, Ar-H), 7.64–7.61 (m, 1H, Ar-H), 7.55–7.49 (m, 2H, Ar-H), 7.46–7.33 (m, 7H, Ar-H), 7.08 (d, *J* = 9.2 Hz, 1H, BzNH), 6.25 (m, 1H, H-13), 5.76 (dd, *J* = 1.6, 9.0 Hz, 1H, H-3′), 5.59 (d, *J* = 7.6 Hz, 1H, H-2), 4.89–4.81 (m, 3H, OCH_2_CH_2_OCO, H-5, H-3), 4.68 (d, *J* = 2.0 Hz, 1H, H-2′), 4.57 (dd, *J* = 6.6, 11.9 Hz, 1H, OCH_2_CH_2_OCO), 4.50 (dd, *J* = 6.5, 11.8 Hz, 1H, OCH_2_CH_2_OCO), 4.44 (d, *J* = 8.0 Hz, 1H, H-20a), 4.38–4.32 (m, 2H, OCH_2_CH_2_OCO, H-20b), 4.15–4.10 (m, 1H, H-7a), 4.07–4.01 (m, *J* = 17.2 Hz, 1H, H-7b), 2.56 (s, 3H, CH_3_), 2.55 (s, 3H, CH_3_), 2.41–2.30 (m, 3H, H-6a, H-6b, H-14a), 2.25–2.18 (m, 1H, H-14b), 1.84 (d, *J* = 1.2 Hz, 3H, CH_3_), 1.25 (s, 3H, CH_3_), 1.20 (s, 3H, CH_3_), 0.82 (s, 9H, SiC(CH_3_)_3_), −0.01 (s, 3H, Si(CH_3_)), −0.26 (s, 3H, Si(CH_3_)); ^13^C-NMR (CDCl_3_, 100 MHz) δ: 192.1, 171.1, 169.4, 165.6, 152.5, 151.7, 144.5, 139.8, 138.4, 134.9, 134.5, 133.6, 131.8, 131.0, 130.3, 128.81 (2C), 128.79 (2C), 128.4, 128.0, 127.0 (2C), 126.5 (2C), 89.6, 84.7, 78.7, 77.3, 75.6, 75.3, 70.8, 69.0, 55.7, 43.0, 42.5, 36.4, 29.7, 26.3, 25.6 (3C), 22.8, 21.6, 18.2, 14.4, 13.6, −5.2, −5.8; LC-MS (ESI, *m/z*): [M + 1], found 1030.4, [M + 23], found 1052.4, C_54_H_64_ClNO_15_Si: 1029.37.

*2′-O-(tert-Butyldimethylsilyl)-2-debenzoyl-2-(3-trifluoromethylbenzoyl)-macrocyclic taxoid* (**19e**). White solid; yield 30.5%; ^1^H-NMR (CDCl_3_, 400 MHz) δ: 8.51 (s, 1H, Ar-H), 8.41 (d, *J* = 7.8 Hz, 1H, Ar-H), 7.92 (d, *J* = 7.8 Hz, 1H, Ar-H), 7.76–7.72 (m, 3H, Ar-H), 7.55–7.51 (m, 1H, Ar-H), 7.47–7.40 (m, 4H, Ar-H), 7.37–7.33 (m, 3H, Ar-H), 7.09 (d, *J* = 9.1 Hz, 1H, NH), 6.24 (t, *J* = 8.8 Hz, 1H, H-13), 5.76 (dd, *J* = 1.0, 9.0 Hz, 1H, H-3′), 5.63 (d, *J* = 7.6 Hz, 1H, H-2), 4.91–4.84 (m, 3H, H-3, H-5, OCH_2_CH_2_OCO), 4.67 (d, *J* = 1.8 Hz, 1H, H-2′), 4.58 (dd, *J* = 6.8, 11.8 Hz, 1H, OCH_2_CH_2_OCO), 4.52 (dd, *J* = 6.8, 13.7 Hz, 1H, OCH_2_CH_2_OCO), 4.44 (d, *J* = 8.1 Hz, 1H, H-20a), 4.38–4.33 (m, 2H, OCH_2_CH_2_OCO, H-20b), 4.11–4.18 (m, 1H, H-7a), 4.09–4.02 (m, *J* = 17.2 Hz, 1H, H-7b), 2.55 (s, 3H, 4-OAc), 2.42–2.36 (m, 3H, H-6a, H-6b, H-14a), 2.29–2.20 (m,1H, H-14b), 1.94 (s, 1H, OH), 1.86 (d, *J* = 1.04 Hz, 3H, CH_3_), 1.29 (s, 3H, CH_3_), 1.26 (s, 3H, CH_3_), 1.22 (s, 3H, CH_3_), 0.84 (s, 9H, SiC(CH_3_)_3_), −0.004 (s, 3H, Si(CH_3_)), −0.24 (s, 3H, Si(CH_3_)); ^13^C-NMR (CDCl_3_, 100 MHz) δ: 192.0, 171.1, 169.3, 167.0, 165.4, 152.5, 151.6, 144.5, 139.8, 138.4, 134.4, 134.3, 133.6, 131.8, 131.4 (q, ^2^*J*_C-F_ = 33.2 Hz), 130.3 (q, ^3^*J*_C-F_ = 3.4 Hz), 130.2, 129.7, 128.8, 128.0, 127.0, 126.9 (q, ^3^*J*_C-F_ = 4.3 Hz), 126.5, 123.7 (q, ^1^*J*_C-F_ = 273.8 Hz), 89.6, 84.7, 78.7, 76.0, 75.8, 75.2, 71.5, 70.8, 69.0, 65.9, 55.7, 43.0, 42.4, 36.4, 31.0, 29.0, 26.3, 25.6, 22.7, 21.6, 18.2, 14.4, 13.6, −5.2,−5.8; LC-MS (ESI, *m/z*): [M + 1], found 1064.4, [M + 23], found 1086.4, C_55_H_64_F_3_NO_15_Si: 1063.40.

#### 3.1.7. General Procedure for the Syntheses of Compounds **21a**–**e**


##### Macrocyclic taxoid (**21a**)

To a solution of of **19a** (32.3 mg, 0.0464 mmol) in THF (8 mL) was added, dropwise, 1.53 mL of HF-pyridine (*v/v* = 1:2) at 0 °C, and the reaction mixture was stirred at room temperature for 30 h. The reaction was quenched with saturated aqueous NaHCO_3_, diluted with ethyl acetate, and washed with saturated NH_4_Cl. The organic layer was dried over anhydrous Na_2_SO_4_ and then concentrated under reduced pressure. Purification of the crude product by silica gel chromatography (ethyl acetate: petroleum ether = 2:1) gave product **21a** as a white solid (13.2 mg, 46.2% yield). ^1^H-NMR (CDCl_3_, 400 MHz) δ: 8.14 (d, *J* = 7.0 Hz, 2H, Ar-H), 7.73 (d, *J* = 8.6 Hz, 2H, Ar-H), 7.62 (t, *J* = 7.5 Hz, 1H, Ar-H), 7.52–7.45 (m, 5H, Ar-H), 7.40–7.30 (m, 5H, Ar-H), 7.22 (d, *J* = 9.1 Hz, 1H, NH), 6.20 (t, *J* = 8.5 Hz, 1H, H-13), 5.78 (dd, *J* = 2.5, 9.1 Hz, 1H, H-3′), 5.58 (d, *J* = 7.5 Hz, 1H, H-2), 4.80–4.74 (overlap, 4H, H-5, H-2′, H-3, OCH_2_CH_2_OCO), 4.52 (dd, *J* = 6.4, 12.0 Hz, 1H, OCH_2_CH_2_OCO), 4.45–4.44 (overlap, 2H, H-20a, OCH_2_CH_2_OCO), 4.31-4.25 (overlap, 2H, H-20b, OCH_2_CH_2_OCO), 4.07–3.91 (overlap, 2H, H-7b, H-7a), 2.56 (br, 1H, OH-2′), 2.51 (s, 3H, CH_3_), 2.35 (s, 3H, CH_3_), 2.33–2.25 (overlap, 4H, H-14a, H-14b, H-6a, H-6b), 1.68 (s, 3H, CH_3_), 1.18 (s, 3H, CH_3_), 1.16 (s, 3H, CH_3_); ^13^C-NMR (CDCl_3_, 150 MHz) δ: 191.8, 172.4, 169.7, 166.9, 166.8, 152.4, 151.5, 144.8, 138.6, 137.9, 135.0, 133.9, 133.6, 132.0, 130.1 (2C), 129.0 (2C), 128.8 (4C), 128.3, 127.0 (2C), 89.4, 84.7, 78.4, 76.1, 75.1, 73.1, 71.8, 71.5, 68.9, 65.8, 54.8, 42.8, 42.4, 36.4, 30.9, 29.7, 26.3, 22.5, 21.2, 14.0, 13.7; LC-MS (ESI, *m/z*): [M + 1], found 882.8, [M + 23], found 904.4, C_48_H_51_NO_15_: 881.33.

*2-Debenzoyl-2-(3-methoxybenzoyl)-macrocyclic taxoid* (**21b**). White solid; yield 54.1%; ^1^H-NMR (CDCl_3_, 400 MHz) δ: 7.76–7.74 (m, 3H, Ar-H), 7.66-7.65 (m, 1H, Ar-H), 7.53–7.46 (m, 8H, Ar-H), 7.36–7.32 (m, 1H, Ar-H), 7.03 (d, *J* = 8.9 Hz, 1H, NH), 6.17 (t, *J* = 8.2 Hz, 1H, H-13), 5.79 (dd, *J* = 2.3, 9.0 Hz, 1H, H-3′), 5.59 (d, *J* = 7.5 Hz, 1H, H-2), 4.84–4.77 (overlap, 4H, H-5, H-2′, H-3, OCH_2_CH_2_OCO), 4.55 (dd, *J* = 6.5, 11.9 Hz, 1H, OCH_2_CH_2_OCO), 4.45 (dd, *J* = 6.6, 13.2 Hz, 1H, OCH_2_CH_2_OCO), 4.43 (d, *J* = 7.5 Hz, 1H, H-20a), 4.36 (d, *J* = 8.29 Hz, 1H, H-20b), 4.32 (dd, *J* = 5.2, 13.8 Hz, 1H, OCH_2_CH_2_OCO), 4.12–3.96 (m, 2H, H-7b, H-7a), 3.88 (s, 3H, OAc), 3.62 (brs, 1H, OH-2′), 2.53 (s, 3H, CH_3_), 2.36 (s, 3H, CH_3_), 2.40–2.27 (overlap, 4H, H-14a, H-14b, H-6a, H-6b), 1.93 (brs, 1H, OH-1), 1.68 (d, 3H, CH_3_), 1.23 (s, 3H, CH_3_), 1.18 (s, 3H, CH_3_); ^13^C-NMR (CDCl_3_, 100 MHz) δ: 191.9, 172.4, 169.7, 167.0, 166.7, 159.8, 152.4, 151.5, 144.8, 138.6, 138.0, 135.1, 133.7, 132.0, 130.3 (2C), 130.0, 129.8, 129.7, 129.0 (2C), 128.7 (4C), 128.3, 127.0 (2C), 122.5, 120.0, 115.0, 89.4, 84.7, 78.4, 76.2, 75.3, 73.2, 71.7, 71.5, 68.9, 65.8, 55.5, 54.9, 42.8, 42.4, 36.4, 31.9, 30.9, 29.7, 26.3, 22.5, 21.2, 14.0, 13.7; LC-MS (ESI, *m/z*): [M + 1], found 912.3, [M + 23], found 934.3, C_49_H_53_NO_16_: 911.34.

*2-Debenzoyl-2-(3-fluorobenzoyl)-macrocyclic taxoid* (**21c**). White solid; yield 52.2%; ^1^H-NMR (CDCl_3_, 400 MHz) δ: 7.97–7.92 (m, 1H, Ar-H), 7.86–7.83 (ddd, *J* = 1.5, 2.5, 9.2 Hz, 1H, Ar-H), 7.75–7.72 (m, 2H, Ar-H), 7.54–7.47 (m, 4H, Ar-H), 7.43–7.39 (m, 4 H, Ar-H), 7.37–7.32 (m, 2H, Ar-H), 7.02 (d, *J* = 9.1 Hz, 1H, NH), 6.17 (dd, *J* = 7.5, 8.8 Hz, 1H, H-13), 5.79 (dd, *J* = 2.4, 9.0 Hz, 1H, H-3′), 5.57 (d, *J* = 7.6 Hz, 1H, H-2), 4.84–4.78 (overlap, 4H, H-5, H-2′, H-3, OCH_2_CH_2_OCO), 4.55 (dd, *J* = 6.4, 11.9 Hz, 1H, OCH_2_CH_2_OCO), 4.47–4.40 (m, 2H, H-20b, H-20a), 4.34–4.29 (m, 2H, OCH_2_CH_2_OCO), 4.12–3.96 (m, 2H, H-7b, H-7a), 3.62 (d, *J* = 4.9 Hz, 1H, OH-2′), 2.53 (s, 3H, CH_3_), 2.36 (s, 3H, CH_3_), 2.35–2.25 (m, 4H, H-14a, H-14b, H-6a, H-6b), 1.93 (brs, 1H, OH-1), 1.68 (d, *J* = 1.2 Hz, 3H, CH_3_), 1.23 (s, 3H, CH_3_), 1.18 (s, 3H, CH_3_); ^13^C-NMR (CDCl_3_, 100 MHz) δ: 191.8, 172.4, 169.6, 167.1, 165.6 (d, *J*_C-F_ = 3.2 Hz, CO), 162.4 (d, ^1^*J*_C-F_ = 247.8 Hz), 152.4, 151.6, 144.7, 138.7, 138.0, 134.8, 133.7, 132.0, 131.3 (d, ^3^*J*_C-F_ = 7.6 Hz), 130.5 (d, ^3^*J*_C-F_ = 7.8 Hz), 129.9 (d, ^4^*J*_C-F_ = 3.9 Hz), 129.7, 129.0, 128.7, 128.4, 127.0, 126.0, 121.0 (d, ^2^*J*_C-F_ = 21.3 Hz), 117.0 (d, ^2^*J*_C-F_ = 23.3 Hz), 89.5, 84.7, 78.5, 76.0, 75.5, 73.2, 71.7, 71.5, 68.9, 65.7, 55.0, 42.8, 42.4, 36.3, 30.9, 26.3, 25.5, 22.7, 22.4, 21.2, 14.1, 14.0, 13.6; LC-MS (ESI, *m/z*): [M + 1], found 900.0, [M + 23], found 922.0, C_48_H_50_FNO_15_: 899.32.

*2-Debenzoyl-2-(3-chlorobenzoyl)-macrocyclic taxoid* (**21d**). White solid; yield 48.7%; ^1^H-NMR (CDCl_3_, 400 MHz) δ: 8.16 (t, 1H, Ar-H), 8.06–8.03 (m, 1H, Ar-H), 7.75–7.72 (m, 2H, Ar-H), 7.61 (ddd, *J* = 1.1, 2.1, 8.0 Hz, 1H, Ar-H), 7.53–7.45 (m, 4H, Ar-H), 7.43–7.39 (m, 4 H, Ar-H), 7.36–7.32 (m, 1H, Ar-H), 7.01 (d, *J* = 9.1 Hz, 1H, NH), 6.17 (td, *J* =1.3, 8.7 Hz, 1H, H-13), 5.78 (dd, *J* = 2.3, 9.0 Hz, 1H, H-3′), 5.56 (d, *J* = 7.6 Hz, 1H, H-2), 4.84–4.78 (overlap,4H, H-5, H-2′, H-3, OCH_2_CH_2_OCO), 4.55 (dd, *J* = 6.4, 11.9 Hz, 1H, OCH_2_CH_2_OCO), 4.47–4.38 (m, 2 H, H-20b, H20a), 4.34–4.29 (m, 2H, OCH_2_CH_2_OCO), 4.11–3.96 (m, 2H, H-7b, H-7a), 3.62 (brs, 1H, OH-2′), 2.52 (s, 3H, CH_3_), 2.37 (s, 3H, CH_3_), 2.42–2.27 (overlap, 4H, H-14a, H-14b, H-6a, H-6b), 2.0 (brs, 1H, OH-1), 1.68 (d, *J* = 1.2 Hz, 3H, CH_3_), 1.23 (s, 3H, CH_3_), 1.18 (s, 3H, CH_3_); ^13^C-NMR (CDCl_3_, 100 MHz) δ: 191.8, 172.4, 169.6, 167.1, 165.5, 152.4, 151.6, 144.7, 138.7, 138.0, 134.8 (2C), 133.8, 133.7, 132.0, 130.9, 130.2 (2C), 129.0, 128.7, 128.4 (2C), 127.1, 127.0, 89.4, 84.7, 78.4, 76.0, 75.5, 73.1, 71.7, 71.5, 68.9, 65.7, 55.0, 42.8, 42.4, 36.3, 30.9, 26.2, 22.4, 21.2, 14.0, 13.6; LC-MS (ESI, *m/z*): [M + 1], found 916.3, [M + 23], found 938.3, C_48_H_50_ClNO_15_: 915.29.

*2-Debenzoyl-2-(3-trifluoromethylbenzoyl)-macrocyclic taxoid* (**21e**). White solid; yield 63.7%; ^1^H-NMR (CDCl_3_, 400 MHz) δ: 8.47 (s, 1H, Ar-H), 8.36 (d, *J* = 7.9 Hz, 1H, Ar-H), 7.90 (d, *J* = 7.8 Hz, 1H, Ar-H), 7.73–7.67 (m, 3H, Ar-H), 7.46–7.52 (m, 3 H, Ar-H), 7.43–7.32 (m, 5H, Ar-H), 6.94 (d, *J* = 9.0 Hz, 1H, NH), 6.16 (dd, *J* = 7.5, 8.8 Hz, 1H, H-13), 5.77 (dd, *J* = 2.0, 8.9 Hz, 1H, H-3′), 5.60 (d, *J* = 7.5 Hz, 1H, H-2), 4.85–4.78 (overlap, 4H, H-5, H-2′, H-3, OCH_2_CH_2_OCO), 4.55 (dd, *J* = 6.6, 12.0 Hz, 1H, OCH_2_CH_2_OCO), 4.45 (dd, *J* = 6.7, 13.6 Hz, 1H, OCH_2_CH_2_OCO), 4.39 (d, *J* = 8.0 Hz, 1 H, H-20b), 4.33 (dd, 1H, OCH_2_CH_2_OCO), 4.31 (d, *J* = 7.7 Hz, 1H, H-20a),4.12-3.97 (m, 2H, H-7b, H-7a), 3.53 (d, *J* = 6.9 Hz, 1H, OH-2′), 2.53 (s, 3H, CH_3_), 2.34 (s, 3H, CH_3_), 2.46–2.28 (m, 4H, H-14a, H-14b, H-6a, H-6b), 1.93 (brs, 1H, OH-1), 1.70 (d, *J* = 1.2 Hz, 3H, CH_3_), 1.24 (s, 3H, CH_3_),1.15 (s, 3H, CH_3_); ^13^C-NMR (CDCl_3_, 100 MHz) δ: 192.0, 171.1, 169.3, 167.0, 165.4, 152.5, 151.6, 144.5, 139.8, 138.4, 134.4, 134.3, 133.6, 131.8, 131.4 (q, ^2^*J*_C-F_ = 33.2 Hz), 130.3 (q, ^3^*J*_C-F_ = 3.4 Hz), 130.2, 129.7, 128.8, 128.0, 127.0, 126.9 (q, ^3^*J*_C-F_ = 4.3 Hz), 126.5, 123.7 (q, ^1^*J*_C-F_ = 273.8 Hz), 89.6, 84.7, 78.7, 76.0, 75.8, 75.2, 71.5, 70.8, 69.0, 65.9, 55.7, 43.0, 42.4, 36.4, 31.0, 29.0, 26.3, 25.6, 22.7, 21.6, 18.2, 14.4, 13.6, −5.2, −5.8; LC-MS (ESI, *m/z*): [M + 1], found 950.3, [M + 23], found 972.3, C_49_H_50_F_3_NO_15_: 949.31. 

#### 3.1.8. General Experimental Procedure for Compounds **18a**–**e**

*2′-O-(tert-Butyldimethylsilyl)-9-O-(3-hydroxyethyl)-10-dehydro-7, 8-seco-10-deacetylpaclitaxel* (**18a**). To a mixture of **15a** (65 mg, 0.0703 mmol), potassium carbonate (67.8 mg, 0.491 mmol) and potassium iodide (11.7 mg, 0.0705 mmol) in DMF (0.65 mL) was added 3-bromo-1-propanol (44 μL 0.487 mmol), dropwise, at 0 °C. The reaction mixture was stirred for 4 h at room temperature and then was diluted with ethyl acetate. The organic phase was washed with saturated aqueous NH_4_Cl. The organic layer was dried over anhydrous Na_2_SO_4_ and then concentrated under reduced pressure. Purification of the crude product by silica gel chromatography (ethyl acetate: petroleum ether = 1:1) gave product **18a** as a white solid (50 mg, 72.4% yield). ^1^H-NMR (CDCl_3_, 400 MHz) δ: 8.10 (d, *J* = 7.6 Hz, 2H, Ar-H), 7.62 (d, *J* = 7.6 Hz, 2H, Ar-H), 7.58 (t, *J* = 7.2 Hz, 1H, Ar-H), 7.47 (dd, *J* = 7.6, 14.8 Hz, 2H, Ar-H), 7.43–7.40 (m, 1H, Ar-H), 7.38–7.36 (m, 4H, Ar-H), 7.30–7.29 (m, 3H, Ar-H), 6.30 (t, *J* = 8.4 Hz, 1H, H-13), 5.88 (d, *J* = 9.6 Hz, 1H, H-3′), 5.57 (d, *J* = 9.6 Hz, 1H, H-2), 5.30–5.27 (overlap, 2H, H-5, H-20a), 4.69 (d, *J* = 2.0 Hz, 1H, H-2′), 4.26 (d, *J* = 7.6 Hz, 1H, H-3), 4.20 (brs, 1H, H-20b), 4.00-3.95 (m, 2H, OCH_2_CH_2_CH_2_OCO), 3.91–3.83 (m, 3H, OCH_2_CH_2_CH_2_OCO, H-7a), 3.76 (brs, 1H, H-7b), 2.75–2.64 (m, 2H, H-6a, H-14a), 2.27–2.21 (m, 2H, H-6b, H-14b), 1.92–1.90 (m, 5H, CH_3_, OCH_2_CH_2_CH_2_OCO), 1.96 (s, 3H, CH_3_), 1.81 (brs, 3H, CH_3_), 1.27 (s, 3H, CH_3_), 1.18 (s, 3H, CH_3_), 0.76 (s, 9H, SiC(CH_3_)_3_), −0.08 (s, 3H, Si(CH_3_)), −0.33 (s, 3H, Si(CH_3_)); ^13^C-NMR (CDCl_3_, 100 MHz) δ: 191.6, 170.7, 168.7, 167.3 (2C), 153.1, 152.8, 144.1, 137.9, 133.8, 133.7, 131.9, 129.9 (2C), 128.9 (2C), 128.7 (4C), 128.6, 127.9 (2C), 126.5 (2C), 82.1, 79.9, 75.7, 74.7, 70.5, 68.8, 60.4, 59.3, 58.8, 55.6, 42.8, 36.6, 32.2, 25.4 (3C), 23.2, 22.1, 18.2, 14.9, 14.2, −5.5, −6.1; LC-MS (ESI, *m/z*): [M + 23], found 1006.0, C_54_H_69_NO_14_Si: 983.45.

*2′-O-(tert-Butyldimethylsilyl)-2-debenzoyl-2-(3-methoxybenzoyl)-9-O-(3-hydroxyethyl)-10-dehydro-7,8-seco-10-deacetylpaclitaxel* (**18b**). White solid; yield 75.3%; ^1^H-NMR (CDCl_3_, 400 MHz) δ: 7.71–7.65 (m, 3H, Ar-H), 7.53 (m, 1H, Ar-H), 7.46–7.41 (m, 2H, Ar-H), 7.11 (dd, *J* = 2.4, 8.0 Hz, 1H, BzNH), 6.28 (t, *J* = 8.6 Hz, 1H, H-13), 5.87 (d, *J* = 9.2 Hz, 1H, H-3′), 5.58 (d, *J* = 9.5 Hz, 1H, H-2), 5.29–5.26 (overlap, 2H, H-5, H-20a), 4.72 (d, *J* = 2.1 Hz, 1H, H-2′), 4.26 (d, *J* = 6.6 Hz, 1H, H-3), 4.21 (brs, 1H, H-20b), 4.00-3.95 (m, 2H, OCH_2_CH_2_CH_2_O), 3.98–3.86 (m, 3H, OCH_2_CH_2_CH_2_O, H-7b), 3.83 (overlap, 4H, OCH_3_, H-7a), 2.77–2.66 (m, 2H, H-6b, H-14b), 2.27–2.21 (m, 2H, H-6a, H-14a), 1.96 (s, 3H, CH_3_), 1.92 (brs, 2H, OCH_2_CH_2_CH_2_O), 1.83 (m, 2H, OH), 1.84 (brs, 3H, CH_3_), 1.26 (s, 3H, CH_3_), 1.19 (s, 3H, CH_3_), 0.78 (s, 9H, SiC(CH_3_)_3_), −0.09 (s, 3H, Si(CH_3_)), −0.29 (s, 3H, Si(CH_3_)); ^13^C-NMR (CDCl_3_, 100 MHz) δ: 191.6, 171.2, 170.9, 168.9, 167.3, 167.2, 159.8, 144.4, 138.1, 133.8, 131.9, 130.4, 130.3, 128.7 (2C), 128.7 (2C), 128.0, 127.0 (2C), 126.7 (2C), 122.3, 119.6, 114.9, 85.8, 79.9, 75.8, 74.9, 70.7, 68.8, 59.3, 59.0, 55.7, 55.5, 42.8, 36.6, 32.3, 25.5 (3C), 25.2, 23.2, 22.2, 18.2, 14.9, 14.5, −5.5, −6.0; LC-MS (ESI, *m/z*): [M + 1], found 1014.4, [M + 23], found 1036.4, C_55_H_71_NO_16_Si: 1013.46.

*2′-O-(tert-Butyldimethylsilyl)-2-debenzoyl-2-(3-fluorobenzoyl)-9-O-(3-hydroxyethyl)-10-dehydro-7, 8-seco-10-deacetylpaclitaxel* (**18c**). White solid; yield 69.7%; ^1^H-NMR (CDCl_3_, 400 MHz) δ: 7.93 (d, *J* = 7.6 Hz, 1H, Ar-H), 7.75 (d, *J* = 8.9 Hz, 1H, Ar-H), 7.63 (d, *J* = 7.5 Hz, 1H, Ar-H), 7.52–7.28 (m, 11H, NH, Ar-H), 6.29 (t, *J* = 8.1 Hz, 1H, H-13), 5.89 (d, *J* = 9.2 Hz, 1H, H-3′), 5.57 (d, *J* = 9.6 Hz, 1H, H-2), 5.28–5.26 (overlap, 2H, H-5, H-20a), 4.70 (d, *J* = 2.0 Hz, 1H, H-2′), 4.27 (d, *J* = 8.1 Hz, 1H, H-3), 4.19 (brs, 1H, H-20b), 3.99–3.94 (m, 2H, OCH_2_CH_2_CH_2_OH), 3.88–3.81 (m, 3H, OCH_2_CH_2_CH_2_OH, H-7a), 3.76 (brs, 1H, H-7b), 2.72–2.63 (m, 2H, H-6a, H-14a), 2.24 (dd, 1H, *J* = 9.5, 15.8 Hz, H-14b), 2.06– 1.98 (overlap, 1 H, H-6b), 1.95 (s, 3H, CH_3_), 1.93–1.87 (m, 4H, CH_3_, OCH_2_CH_2_CH_2_OH), 1.83 (brs, 3H, CH_3_), 1.26 (brs, 3H, CH_3_), 1.17 (s, 3H, CH_3_), 0.77 (s, 9H, SiC(CH_3_)_3_), −0.09 (s, 3H, Si(CH_3_)), −0.32 (s, 3H, Si(CH_3_)); ^13^C-NMR (CDCl_3_, 100 MHz) δ: 191.5, 171.2, 170.8, 168.6, 167.4, 166.3, 162.7 (d, ^1^J_C-F_ = 248.2 Hz), 144.3, 137.9, 133.9, 131.9, 131.2 (d, ^2^J_C-F_ = 7.3 Hz), 131.0 (d, ^3^J_C-F_ = 7.7 Hz), 128.8, 128.7, 128.0, 127.0, 126.5, 125.8 (d, ^4^J_C-F_ = 2.6 Hz), 121.0 (d, ^2^J_C-F_ = 21.2 Hz), 116.6 (d, ^2^J_C-F_ = 22.8 Hz), 85.9, 80.0, 75.8, 75.2, 70.5, 68.9, 59.3, 55.6, 42.8, 36.7, 32.3, 25.5, 23.2, 22.1, 18.2, 15.0, 14.5, −5.4, −6.0; LC-MS (ESI, *m/z*): [M + 1], found 1002.0, [M + 23], found 1025.0, C_54_H_68_FNO_14_Si: 1001.44.

*2′-O-(tert-Butyldimethylsilyl)-2-debenzoyl-2-(3-chlorobenzoyl)-9-O-(3-hydroxyethyl)-10-dehydro-7,8-seco-10-deacetylpaclitaxel* (**18d**). White solid; yield 52.6%; ^1^H-NMR (CDCl_3_, 400 MHz) δ: 8.03 (d, *J* = 7.6 Hz, 1H, Ar-H), 8.00–7.96 (m, *J* = 7.6 Hz, 1H, Ar-H), 7.65 (d, *J* = 7.6 Hz, 2H, Ar-H), 7.54 (t, *J* = 7.2 Hz, 1H, Ar-H), 7.48–7.44 (m, 2H, Ar-H), 7.43–7.40 (m, 1H, Ar-H), 7.38–7.36 (m, 4H, Ar-H), 7.39–7.28 (m, 7H, Ar-H), 6.30 (t, *J* = 8.4 Hz, 1H, H-13), 5.88 (d, *J* = 9.6 Hz, 1H, H-3′), 5.57 (d, *J* = 9.6 Hz, 1H, H-2), 5.30–5.27 (overlap, 2H, H-5, H-20a), 4.71 (s, 1H, H-2′), 4.26 (d, *J* = 7.6 Hz, 1H, H-3), 4.18 (brs, 1H, H-20b), 3.99–3.93 (m, 2H, OCH_2_CH_2_CH_2_O), 3.79–3.40 (m, 3H, OCH_2_CH_2_CH_2_O, H-7a), 3.73–3.64 (brs, 1H, H-7b), 2.94 (s, 3H, CH_3_), 2.84 (s, 3H, CH_3_), 2.66 (br, 2H, H-6a, H-14a), 2.29–2.22 (m, 2H, H-6b, H-14b), 1.93 (s, 3H, CH_3_), 1.93–1.86 (m, 5H, CH_3_, OCH_2_CH_2_CH_2_O), 1.82 (brs, 3H, CH_3_), 1.24 (s, 3H, CH_3_), 1.16 (s, 3H, CH_3_), 0.76 (s, 9H, SiC(CH_3_)_3_), −0.11 (s, 3H, Si(CH_3_)), −0.31 (s, 3H, Si(CH_3_)); ^13^C-NMR (CDCl_3_, 100 MHz) δ: 191.5, 171.1, 170.8, 167.4, 165.9, 162.5, 144.3, 137.9, 134.8, 133.9, 133.7, 131.8, 131.0, 130.6, 129.5, 128.7 (2C), 128.6 (2C), 128.1, 127.9, 126.9 (2C), 126.5 (2C), 85.8, 75.7, 75.2, 70.5, 68.8, 67.9, 60.4, 59.2, 55.6, 42.7, 36.6, 36.4, 32.3, 31.4, 25.4 (3C), 23.2, 22.1, 21.0, 18.2, 14.5, 14.2, −5.5, −6.1; LC-MS (ESI, *m/z*): [M + 1], found 1018.4, [M + 23], found 1040.4, C_54_H_68_ClNO_14_Si: 1017.41.

*2′-O-(tert-Butyldimethylsilyl)-2-debenzoyl-2-(3-trifluoromethylbenzoyl)-9-O-(3-hydroxyethyl)-10-dehydro-7, 8-seco-10-deacetylpaclitaxel* (**18e**). White solid; yield 71.8%; ^1^H-NMR (CDCl_3_, 400 MHz): δ 8.38 (d, *J* = 7.8 Hz, 2H, Ar-H), 7.83 (d, *J* = 7.7 Hz, 2H, Ar-H), 7.68 (t, *J* = 7.8 Hz, 1H, Ar-H), 7.44 (dd, *J* = 15.4, 8.0 Hz, 2H, Ar-H), 7.47–7.41 (m, 1H, Ar-H), 7.40–7.36 (m, 4H, Ar-H), 7.30–7.29 (m, 3H, Ar-H), 6.29 (t, *J* = 8.5 Hz, 1H, H-13), 5.86 (d, *J* = 8.6 Hz, 1H, H-3′), 5.59 (d, *J* = 9.5 Hz, 1H, H-2), 5.26 (overlap, 2H, H-5, H-20a), 4.70 (d, *J* = 2.0 Hz, 1H, H-2′), 4.29 (d, *J* = 7.3 Hz, 1H, H-3), 4.21 (brs, 1H, H-20b), 4.00–3.96 (m, 2H, OCH_2_CH_2_CH_2_OH), 3.90–3.87 (m, 3H, OCH_2_CH_2_CH_2_OH, H-7a), 3.75 (brs, 1H, H-7b), 2.70–2.60 (m, 2H, H-6a, H-14a), 2.28–2.22 (m, 2H, H-6b, H-14b), 1.97 (s, 3H, CH_3_), 1.94–1.92 (m, 5H, CH_3_, OCH_2_CH_2_CH_2_OH), 1.81 (brs, 3H, CH_3_), 1.27 (s, 3H, CH_3_), 1.19 (s, 3H, CH_3_), 0.76 (s, 9H, SiC(CH_3_)_3_), −0.08 (s, 3H, Si(CH_3_)), −0.32 (s, 3H, Si(CH_3_)); ^13^C-NMR (CDCl_3_, 100 MHz) δ: 191.5, 170.8, 167.5, 144.3, 137.9, 133.9, 133.2, 132.0, 131.4 (q, ^2^J_C-F_ = 33.3 Hz), 130.2 (q, ^3^J_C-F_ = 4.0 Hz), 130.0, 128.8, 128.1, 126.9, 126.8 (q, ^3^J_C-F_ = 3.9 Hz), 126.5, 123.6 (q, ^1^J_C-F_ = 274.2 Hz), 85.9, 75.6, 75.4, 70.5, 59.4, 55.6, 42.9, 36.7, 32.3, 25.5, 18.2, 14.6, −5.4, −6.0; LC-MS (ESI, *m/z*): [M + 1], found 1052.4, [M + 23], found 1074.4, C_55_H_68_NO_14_F_3_Si: 1051.44.

#### 3.1.9. General Experimental Procedure for Compounds **20a**–**e**

*2′-O-(tert-Butyldimethylsilyl)-macrocyclic taxoid* (**20a**). To a solution of **18a** (26.2 mg, 0.0266 mmol) in dichloromethane (5.7 mL) was added, dropwise, pyridine (11 μL, 0.133 mmol) and triphosgene (8.7 mg, 0.0293 mmol) at 0 °C, then the reaction mixture was stirred at room temperature for 3.5 h. The reaction was diluted with ethyl acetate, then washed with saturated NaCl. The organic layer was dried over anhydrous Na_2_SO_4_ and then concentrated under reduced pressure. Purification of the crude product by silica gel chromatography (ethyl acetate: petroleum ether = 1:3) gave product **20a** as a white solid (6.7 mg, 24.9% yield). ^1^H-NMR (CDCl_3_, 400 MHz) δ: 8.17 (d, *J* = 8.0 Hz, 2H, Ar-H), 7.75 (d, *J* = 7.6 Hz, 2H, Ar-H), 7.63 (t, *J* = 7.2 Hz, 1H, Ar-H), 7.56–7.49 (m, 3H, Ar-H), 7.44–7.29 (m, 7H, Ar-H), 7.09 (d, *J* = 9.2 Hz, 1H, NH), 6.27 (t, *J* = 8.8 Hz, 1H, H-13), 5.76 (d, *J* = 8.8 Hz, 1H, H-3′), 5.60 (d, *J* = 8.0 Hz, 1H, H-2), 4.80 (d, *J* = 7.6 Hz, 2H, H-5, H-3), 4.67 (brs, 1H, H-2′), 4.63 (m, 1H, OCH_2_CH_2_CH_2_OCO), 4.53 (dd, *J* = 6.4, 11.6 Hz, 1H, OCH_2_CH_2_CH_2_OCO), 4.47 (d, *J* = 8.4 Hz, 1H, H-20a), 4.42 (dd, *J* = 4.0, 7.2 Hz, 1H, OCH_2_CH_2_CH_2_OCO), 4.34 (d, *J* = 8.4 Hz, 1H, H-20b), 4.11 (overlap, 2H, H-7a, OCH_2_CH_2_CH_2_OCO), 3.98 (dt, *J* = 6.0, 11.6 Hz, 1H, H-7b), 2.66 (m, 1H, H-6a), 2.56 (s, 3H, CH_3_), 2.52 (s, 3H, CH_3_), 2.42–2.11 (m, 5H, H-6b, H-14a, H-14b, OCH_2_CH_2_CH_2_O), 1.82 (s, 3H, CH_3_), 1.24 (s, 3H, CH_3_), 1.17 (s, 3H, CH_3_), 0.79 (s, 9H, SiC(CH_3_)_3_), −0.04 (s, 3H, Si(CH_3_)), −0.31 (s, 3H, Si(CH_3_)); ^13^C-NMR (CDCl_3_, 150 MHz) δ: 191.6, 170.7, 169.7, 169.3, 166.7, 166.6, 153.6, 152.4, 144.1, 138.8, 138.1, 135.5, 133.6, 131.6, 130.2 (2C), 130.0, 128.7, 128.6, 128.5, 127.7, 126.8 (2C), 126.1 (2C), 90.3, 84.4, 78.4, 76.1, 75.2, 75.0, 70.4, 68.9, 65.8, 64.2, 55.4, 42.6 (2C), 36.4, 30.8, 28.8, 25.9, 25.3 (3C), 22.5, 21.5, 17.9, 14.5, 14.1, −5.5, −6.1; LC-MS (ESI, *m/z*): [M + 23], found 1032.0, C_55_H_67_NO_15_Si: 1009.43. 

*2′-O-(tert-Butyldimethylsilyl)-2-debenzoyl-2-(3-methoxybenzoyl)-macrocyclic taxoid* (**20b**). White solid; yield 28.6%; ^1^H-NMR (CDCl_3_, 400MHz) δ: 7.79 (d, *J* = 7.6 Hz, 1H, Ar-H), 7.76 (d, *J* = 7.2 Hz, 2H, Ar-H), 7.67–7.66 (m, 1H, Ar-H), 7.55–7.50 (m, 1H, Ar-H), 7.48–7.42 (m, 3H, Ar-H), 7.40–7.29 (m, 5H, Ar-H), 7.19–7.16 (m, 1H, Ar-H), 7.09 (d, *J* = 8.8 Hz, 1H, BzNH), 6.27 (t, *J* = 8.8 Hz, 1H, H-13), 5.76 (dd, *J* = 1.5, 9.0 Hz, 1H, H-3′), 5.62 (d, *J* = 7.64 Hz, 1H, H-2), 4.85–4.82 (m, 2H, H-5, H-3), 4.69–4.63 (m, 2H, H-2′, OCH_2_CH_2_CH_2_OCO), 4.56–4.51 (m, 1H, OCH_2_CH_2_CH_2_OCO), 4.48–4.38 (m, 3H, H-20a, H-20b, OCH_2_CH_2_CH_2_OCO), 4.16–4.07 (m, 2H, H-7a, OCH_2_CH_2_CH_2_OCO), 4.01–3.96 (m, 1H, H-7b), 3.90 (s, 3H, OCH_3_), 2.72–2.62 (m, 1H, H-6a), 2.57 (s, 3H, CH_3_), 2.53 (s, 3H, CH_3_), 2.43–2.36 (m, 1H, H-6b), 2.35–2.28 (m, 1H, H-14a), 2.16–2.14 (m, 2H, H-14b, OCH_2_CH_2_CH_2_O), 1.82 (d, *J* = 1.1 Hz, 3H, CH_3_), 1.25 (s, 3H, CH_3_), 1.18 (s, 3H, CH_3_), 0.88 (s, 9H, SiC(CH_3_)_3_), −0.014 (s, 3H, Si(CH_3_)), −0.29 (s, 3H, Si(CH_3_)); ^13^C-NMR (CDCl_3_, 100 MHz) δ: 190.8, 171.3, 171.2, 169.2, 166.9, 166.8, 159.8, 144.8, 138.3, 134.2, 131.8, 130.6, 130.1, 128.8 (2C), 128.7 (2C), 128.0, 127.0 (2C), 126.6 (2C), 122.3, 120.0, 114.5, 85.1, 77.4, 77.3, 76.8, 75.3, 74.9, 70.8, 67.8, 60.4, 55.8, 55.5, 42.8, 41.6, 40.4, 36.6, 33.0, 29.7, 25.5 (3C), 23.2, 22.4, 21.0, 18.2, 14.5, 14.2, −5.3,−5.8; LC-MS (ESI, *m/z*): [M + 1], found 1040.5, [M + 23], found 1062.5, C_56_H_69_NO_16_Si: 1039.44.

*2′-O-(tert-Butyldimethylsilyl)-2-debenzoyl-2-(3-fluorobenzoyl)-macrocyclic taxoid* (**20c**). White solid; yield 27.1%; ^1^H-NMR (CDCl_3_, 400 MHz) δ: 8.00 (d, *J* = 7.7 Hz, 1H, Ar-H), 7.87 (d, *J* = 8.8 Hz, 1H, Ar-H), 7.77 (d, *J* = 7.32 Hz, 1H, Ar-H), 7.57–7.51 (m, 2H, Ar-H), 7.46–7.33 (m, 8H, Ar-H), 7.10 (d, *J* = 9.0 Hz, 1H, NH), 6.26 (t, *J* = 8.5 Hz, 1H, H-13), 5.78 (d, *J* = 8.5 Hz, 1H, H-3′), 5.60 (d, *J* = 7.7 Hz, 1H, H-2), 4.85–4.82 (overlap, 2H, H-5, H-3), 4.72–4.63 (overlap, 2H, OCH_2_CH_2_CH_2_OCO, H-2′), 4.55 (dt, *J* = 5.5, 10.3 Hz, 1H, OCH_2_CH_2_CH_2_OCO), 4.47–4.42 (overlap, 2H, OCH_2_CH_2_CH_2_OCO, H-20a), 4.37 (d, *J* = 8.2 Hz, 1H, H-20b), 4.14–4.10 (overlap, 2H, H-7a, OCH_2_CH_2_CH_2_OCO), 3.99 (dt, *J* = 6.4, 11.3 Hz, 1H, H-7b), 2.73–2.64 (m, 1H, H-6a), 2.57 (s, 3H, CH_3_), 2.54 (s, 3H, CH_3_), 2.42–2.18 (m, 5H, H-6b, H-14a, H-14b, OCH_2_CH_2_CH_2_O), 2.10 (brs, 1H, OH), 1.83 (s, 3H, CH_3_), 1.25 (s, 3H, CH_3_), 1.18 (s, 3H, CH_3_), 0.82 (s, 9H, SiC(CH_3_)_3_), −0.008 (s, 3H, Si(CH_3_)), −0.27 (s, 3H, Si(CH_3_)); ^13^C-NMR (CDCl_3_, 100 MHz) δ: 191.8, 171.0, 169.5, 166.9, 165.7 (d, ^5^*J*_C-F_ = 3.0 Hz, ArCO), 162.6 (d, ^1^*J*_C-F_ = 247.8 Hz), 153.8, 152.6, 144.3, 139.1, 138.4, 135.5, 134.2, 131.8, 131.3 (d, ^3^*J*_C-F_ = 7.5 Hz), 130.6 (d, ^3^*J*_C-F_ = 7.7 Hz), 128.8, 128.7, 127.9, 127.0, 126.4, 126.0 (d, ^4^*J*_C-F_ = 2.6 Hz), 120.9 (d, ^2^*J*_C-F_ = 21.1 Hz), 117.0 (d, ^2^*J*_C-F_ = 23.2 Hz), 90.6, 84.6, 78.6, 77.4, 77.1, 76.7, 76.2, 75.8, 75.3, 70.6, 69.2, 66.0, 64.4, 55.6, 42.8 (2C), 36.5, 31.0, 29.0, 26.1, 25.5 (3C), 22.7, 21.7, 18.2, 14.7, 14.4, −5.2, −5.8; LC-MS (ESI, *m/z*): [M + 1], found 1028.4, [M + 23], found 1050.4, C_55_H_66_FNO_15_Si: 1027.42.

*2′-O-(tert-Butyldimethylsilyl)-2-debenzoyl-2-(3-chlorobenzoyl)-macrocyclic taxoid* (**20d**). White solid; yield 23.6%; ^1^H-NMR (CDCl_3_, 400 MHz) δ: 8.20 (t, *J* = 1.6 Hz, 1H, Ar-H), 8.12–8.09 (m, 1H, Ar-H), 7.77 (d, *J* = 7.2 Hz, 1H, Ar-H), 7.65–7.62 (m, 1H, Ar-H), 7.56–7.51 (m, 2H, Ar-H), 7.47–7.33 (m, 7H, Ar-H), 7.09 (d, *J* = 9.0 Hz, 1H, BzNH), 6.26 (t, *J* = 8.8 Hz, 1H, H-13), 5.77 (dd, *J* = 1.5, 8.98 Hz, 1H, H-3′), 5.60 (d, *J* = 7.8 Hz, 1H, H-2), 4.85–4.82 (dd, *J* = 3.5, 11.0 Hz, 2H, H-5, H-17b), 4.72–4.64 (m, 2H, OCH_2_CH_2_CH_2_OCO, H-2′), 4.59–4.53 (m, 1H, OCH_2_CH_2_CH_2_OCO), 4.49–4.43 (m, 2H, OCH_2_CH_2_CH_2_OCO, H-20a), 4.38 (d, *J* = 8.4 Hz, 1H, H-20b), 4.15–4.10 (m, 2H, OCH_2_CH_2_CH_2_OCO, H-20a), 4.02–3.96 (m, 1H, H-7b), 2.74–2.65 (m, 1H, H-6a), 2.57 (s, 3H, CH_3_), 2.55 (s, 3H, CH_3_), 2.42–2.20 (m, 5H, H-6b, H-14a, H-14b, OCH_2_CH_2_CH_2_O), 1.84 (d, *J* = 1.0 Hz, 3H, CH_3_), 1.26 (s, 3H, CH_3_), 1.19 (s, 3H, CH_3_), 0.83 (s, 9H, SiC(CH_3_)_3_), −0.005 (s, 3H, Si(CH_3_)), −0.26 (s, 3H, Si(CH_3_)); ^13^C-NMR (CDCl_3_, 100 MHz) δ: 190.7, 171.2, 169.1, 167.0, 165.7, 144.8, 138.3, 135.5, 135.0, 134.3, 133.9, 131.8, 131.1, 130.5, 129.9, 128.8 (2C), 128.7 (2C), 128.2, 128.0, 127.0 (2C), 126.7 (2C), 90.6, 84.6, 78.7, 76.2, 75.8, 75.3, 70.7, 67.8, 55.7, 42.8, 41.6, 36.5, 34.7, 31.6, 29.7, 25.6 (3C), 22.7, 22.4, 18.2, 14.5, 14.1, −5.3, −5.8; LC-MS (ESI, *m/z*): [M + 1], found 1044.0, [M + 23], found 1065.9, C_55_H_66_ClNO_15_Si: 1043.39.

*2′-O-(tert-Butyldimethylsilyl)-2-debenzoyl-2-(3-trifluoromethylbenzoyl)-macrocyclic taxoid* (**20e**). White solid; yield 26.7%; ^1^H-NMR (CDCl_3_, 400MHz) δ: 8.50 (s, 1H, Ar-H), 8.41 (d, *J* = 7.8 Hz, 1H, Ar-H), 7.92 (d, *J* = 7.9 Hz, 1H, Ar-H), 7.55–7.51 (m, 1H, Ar-H), 7.48–7.33 (m, 8H, Ar-H), 7.09 (d, *J* = 9.0 Hz, 1H, NH), 6.26–6.21 (m, 1H, H-13), 5.76 (dd, *J* =1.4, 9.1 Hz, 1H, H-3′), 5.64 (d, *J* = 7.8 Hz, 1H, H-2), 4.89–4.83 (m, 2H, H-5, H-3), 4.69–4.63 (m, 2H, OCH_2_CH_2_CH_2_OCO, H-2′), 4.60–4.55 (m, 1H, OCH_2_CH_2_CH_2_OCO), 4.49–4.43 (m, 2H, OCH_2_CH_2_CH_2_OCO, H-20a), 4.37 (d, *J* = 8.0 Hz,1H, H-20b), 4.13 (t, *J* = 7.7 Hz, 2H, H-7a, OCH_2_CH_2_CH_2_OCO), 2.75–2.67 (m, 1H, H-6a), 4.02–3.96 (m, 1H, H-7b), 2.58 (s, 3H, CH_3_), 2.53 (s, 3H, CH_3_), 2.45–2.18 (m, 5H, H-6b, H-14a, H-14b, OCH_2_CH_2_CH_2_O), 1.84 (d, *J* =1.0 Hz, 1H, CH_3_), 1.26 (s, 3H, CH_3_), 1.25 (s, 3H, CH_3_), 1.20 (s, 3H, CH_3_), 0.83 (s, 9H, SiC(CH_3_)_3_), −0.006 (s, 3H, Si(CH_3_)), −0.25 (s, 3H, Si(CH_3_)); ^13^C-NMR (CDCl_3_, 100 MHz) δ: 191.7, 171.1, 167.0, 165.5, 153.9, 152.7, 135.4, 134.3, 133.6, 131.8, 131.4 (q, ^2^*J* = 33.3 Hz), 130.9, 130.3 (q, ^3^*J* = 3.9 Hz), 130.2, 129.7, 128.8, 128.0, 127.0, 126.9 (q, ^3^*J* = 3.8 Hz), 126.5, 123.7 (q, ^1^*J* = 273.8 Hz), 90.6, 84.7, 78.7, 76.2, 75.8, 75.3, 70.8, 69.2, 66.0, 64.5, 55.7, 42.9, 42.8, 36.5, 31.0, 29.0, 26.1, 22.6, 21.6, 18.2, 14.6, 14.4, −5.2, −5.8; LC-MS (ESI, *m*/*z*): [M + 1], found 1078.4, [M + 23], found 1100.4, C_56_H_66_F_3_NO_15_Si: 1077.42.

#### 3.1.10. General Procedure for the Syntheses of Compounds **22a**–**e**


##### Macrocyclic taxoid (**22a**)

To a solution of of **20a** (29.6 mg, 0.0293 mmol) in THF (8 mL) was added, dropwise, 0.47 mL of HF-pyridine (*v/v* = 1:2) at 0 °C, and the reaction mixture was stirred at room temperature for 30 h. The reaction was quenched with saturated aqueous NaHCO_3_, diluted with ethyl acetate, and washed with saturated NH_4_Cl. The organic layer was dried over anhydrous Na_2_SO_4_ and then concentrated under reduced pressure. Purification of the crude product by silica gel chromatography (ethyl acetate: petroleum ether = 2:1) gave product **22a** as a white solid (13.4 mg, 51.0% yield). ^1^H-NMR (CDCl_3_, 400 MHz) δ: 7.16 (d, *J* = 7.1 Hz, 2H, Ar-H), 7.75 (d, *J* = 7.0 Hz, 2H, Ar-H), 7.64 (t, *J* = 7.5 Hz, 1H, Ar-H), 7.54–7.50 (m, 3H, Ar-H), 7.49–7.47 (m, 2H, Ar-H), 7.43–7.39 (m, 4H, Ar-H), 7.36–7.32 (m,1H, Ar-H), 7.04 (d, *J* = 9.1 Hz, 1H, NH), 6.18 (dd, *J* = 8.7 Hz, 1H, H-13), 5.81 (dd, *J* = 2.4, 9.1 Hz, 1H, H-3′), 5.59 (d, *J* = 7.7 Hz, 1H, H-2), 4.81–4.79 (overlap, 3H, H-5, H-3, H-2′), 4.61 (dt, *J* = 5.4, 11.0 Hz, 1H, OCH_2_CH_2_CH_2_OCO), 4.52 (dt, *J* = 6.9, 12.0 Hz, 1H, OCH_2_CH_2_CH_2_OCO), 4.44 (overlap, 2H, H-20a, OCH_2_CH_2_CH_2_OCO), 4.41–4.33 (m, 1H, H-20b), 4.34–4.28 (overlap, *J* = 6.7 Hz, H-7a), 4.08 (overlap, 2H, H-7a, OCH_2_CH_2_CH_2_OCO), 3.95 (m, 1H, H-7b), 3.58 (brs, 1H, OH-2′), 2.66–2.61 (m, 1H, H-6a), 2.55 (s, 3H, CH_3_), 2.37–2.33 (brs, 3H, CH_3_), 2.30–2.12 (overlap, 3H, H-6b, H-14a, H-14b), 1.67 (brs, 2H, OCH_2_CH_2_CH_2_OCO), 1.23 (s, 3H, CH_3_), 1.16 (s, 3H, CH_3_); ^13^C-NMR (CDCl_3_, 100 MHz) δ: 191.6, 172.4, 169.8, 167.0, 166.9, 153.8, 152.6, 144.6, 138.0 (2C), 136.1, 133.9, 133.7, 130.2 (2C), 129.1, 129.0 (2C), 128.9, 128.8 (2C), 128.7 (2C), 128.3, 127.0 (2C), 90.5, 84.7, 78.5, 76.3, 75.3, 73.2, 71.7, 69.2, 65.6, 64.4, 54.8, 42.9, 42.7, 36.5, 31.0, 29.0, 26.2, 22.4, 21.3, 14.7, 14.0; LC-MS (ESI, *m/z*): [M + 1], found 896.3, [M + 23], found 918.3, C_49_H_53_NO_15_: 895.34. 

*2-Debenzoyl-2-(3-methoxybenzoyl)-macrocyclic taxoid* (**22b**). White solid; yield 59.2%; ^1^H-NMR (CDCl_3_, 400 MHz) δ: 7.75 (d, *J* = 7.4 Hz, 3H, Ar-H), 7.65 (dd, *J* =1.42, 2.3 Hz, 1H, Ar-H), 7.52–7.38 (m, 9H, Ar-H), 7.17 (dd, *J* = 2.1, 8.2 Hz, 1H, Ar-H), 7.06 (d, *J* = 9.0 Hz, 1H, NH), 6.15 (t, *J* = 8.2 Hz, 1H, H-13), 5.78 (dd, *J* = 2.1, 8.9 Hz, 1H, H-3′), 5.59 (d, *J* = 7.6 Hz, 1H, H-2), 4.80–4.76 (overlap, 3H, H-5, H-3, H-2′), 4.60 (dt, *J* = 5.6, 11.2 Hz, 1H, OCH_2_CH_2_CH_2_OCO), 4.51 (dt, *J* = 5.7, 11.2 Hz, 1H, OCH_2_CH_2_CH_2_OCO), 4.44–4.37 (overlap, *J* = 5.6 Hz, 2H, OCH_2_CH_2_CH_2_OCO, H-20a), 4.36 (d, *J* = 8.3 Hz, 1H, H-20b), 4.07 (overlap, 2H, H-7a, OCH_2_CH_2_CH_2_OCO), 3.97–3.91 (m, 1H, H-7b), 3.87 (s, 3H, OCH_3_), 3.66 (d, *J* = 4.1 Hz, 1H, OH-2′), 2.67–2.56 (m, 1H, H-6a), 2.55 (s, 3H, CH_3_), 2.34 (s, 3H, CH_3_), 2.40–2.30 (m, 2H, OCH_2_CH_2_CH_2_OCO), 2.28–2.12 (m, 3H, H-6b, H-14a, H-14b), 2.02 (brs, 1H, OH-1), 1.65 (s, 3H, CH_3_), 1.22 (s, 3H, CH_3_), 1.15 (s, 3H, CH_3_); ^13^C-NMR (CDCl_3_, 100 MHz) δ: 191.6, 172.3, 169.8, 167.0, 166.8, 159.8, 153.7, 152.6, 144.6, 138.1, 137.9, 136.1, 133.7, 132.0, 130.3, 129.8, 129.0, 128.7, 128.3, 127.1, 127.0, 122.4, 112.0, 115.0, 90.4, 84.7, 78.4, 76.3, 75.3, 73.3, 71.7, 69.2, 66.0, 64.4, 55.5, 54.9, 42.8, 42.7, 36.5, 31.0, 29.0, 26.1, 22.4, 21.2, 14.7, 14.0; LC-MS (ESI, *m/z*): [M + 1], found 926.3, [M + 23], found 948.3, C_50_H_55_NO_16_: 925.35.

*2-Debenzoyl-2-(3-fluorobenzoyl)-macrocyclic taxoid* (**22c**). White solid; yield 51.5%; ^1^H-NMR (CDCl_3_, 400 MHz) δ: 7.96 (d, *J* = 7.69 Hz, 1 H, Ar-H), 7.84 (d, *J* = 9.1 Hz, 1H, Ar-H), 7.74 (d, *J* = 7.7 Hz, 2H, Ar-H), 7.53–7.46 (m, 4 H, Ar-H), 7.40 (t, *J* = 7.5 Hz, 4H, Ar-H), 7.36–7.32 (m, 2H, Ar-H), 7.05 (d, *J* = 9.0 Hz, 1H, NH), 6.15 (t, *J* = 8.6 Hz, 1H, H-13), 5.78 (dd, *J* = 2.0, 9.0 Hz, 1H, H-3′), 5.57 (d, *J* = 7.7 Hz, 1H, H-2), 4.81–4.79 (overlap, 3H, H-5, H-3, H-2′), 4.60 (dt, *J* = 5.3, 10.7 Hz, 1H, OCH_2_CH_2_CH_2_OCO), 4.52 (dt, *J* = 5.4, 10.8 Hz, 1H, OCH_2_CH_2_CH_2_OCO), 4.44–4.40 (ovelap, 2H, H-7a, OCH_2_CH_2_CH_2_OCO), 4.40 (d, *J* = 8.0 Hz, 1H, H-20a), 4.32 (d, *J* = 8.1 Hz, 1H, H-20b), 4.07 (t, *J* = 7.5 Hz, 2H, OCH_2_CH_2_CH_2_OCO), 3.93 (dt, *J* = 5.9, 11.9 Hz, 1H, H-7b), 3.69 (brs, 1H, OH-2′), 2.68–2.58 (m, 1H, H-6a), 2.54 (s, 3H, CH_3_), 2.34 (s, 3H, CH_3_), 2.33–2.26 (m, 2H, OCH_2_CH_2_CH_2_OCO), 2.24–2.16 (m, 3H, H-6b, H-14a, H-14b), 1.97 (brs, 1H, OH-1), 1.65 (br, 3H, CH_3_), 1.22 (s, 3H, CH_3_), 1.15 (s, 3H, CH_3_); ^13^C-NMR (CDCl_3_, 100 MHz) δ: 191.5, 172.4, 169.7, 167.0, 165.7 (d, *J*_C-F_ = 2.8 Hz, CO), 162.6 (d, ^1^*J*_C-F_ =247.9 Hz), 153.8, 152.6, 144.5, 138.1 (2C), 135.9, 133.8, 131.9, 131.3 (d, ^3^*J*_C-F_ = 7.5 Hz), 130.5 (d, ^3^*J*_C-F_ = 7.6 Hz), 129.0, 128.7, 128.3, 127.0 (2C), 126.0 (d, ^4^*J*_C-F_ = 2.9 Hz), 121.0 (d, ^2^*J*_C-F_ = 21.3 Hz), 116.9 (d, ^2^*J*_C-F_ = 23.3 Hz), 90.5, 84.6, 78.5, 76.2, 75.6, 73.3, 71.6, 69.2, 66.0, 64.4, 55.0, 42.8, 42.7, 36.4, 30.9, 29.3, 29.0, 26.1, 22.3, 21.3, 14.6, 14.0; LC-MS (ESI, *m/z*): [M + 1], found 914.3, [M + 23], found 936.3, C_49_H_52_FNO_15_: 913.33.

*2-Debenzoyl-2-(3-chlorobenzoyl)-macrocyclic taxoid* (**22d**). White solid; yield 65.4%; ^1^H-NMR (CDCl_3_, 400 MHz) δ: 8.16 (t, *J* = 1.8 Hz, 1H, Ar), 8.06–8.04 (m, 1 H, Ar-H), 7.75–7.73 (m, 2H, Ar-H), 7.61 (ddd, *J* =1.0, 2.0, 8.0 Hz, 1H, Ar-H), 7.43–7.39 (m, 4H, Ar-H), 7.52–7.45 (m, 4 H, Ar-H), 7.37–7.32 (m, 1H, Ar-H), 7.01 (d, *J* = 9.0 Hz, 1H, NH), 6.15 (dd, *J* = 7.4, 8.7 Hz, 1H, H-13), 5.78 (dd, *J* = 2.3, 9.0 Hz, 1H, H-3′), 5.57 (d, *J* = 7.7 Hz, 1H, H-2), 4.81–4.79 (overlap, 3H, H-5, H-3, H-2′), 4.60 (dt, *J* = 5.3, 10.7 Hz, 1H, OCH_2_CH_2_CH_2_OCO), 4.53 (dt, *J* = 5.5, 11.3 Hz, 1H, OCH_2_CH_2_CH_2_OCO), 4.45–4.44 (m, 1H, OCH_2_CH_2_CH_2_OCO), 4.39 (d, *J* = 8.2 Hz, 1H, H-20a), 4.33 (d, *J* = 8.2 Hz, 1H, H-20b), 4.08 (overlap, *J* = 7.62 Hz, 2H, H-7a, OCH_2_CH_2_CH_2_OCO), 3.96–3.90 (m, 1H, H-7b), 3.60 (brs, 1H, OH-2′), 2.68–2.58 (m, 1H, H-6a), 2.54 (s, 3H, CH_3_), 2.35 (s, 3H, CH_3_), 2.44–2.30 (m, 2H, OCH_2_CH_2_CH_2_OCO), 2.28–2.16 (m, 3H, H-6b, H-14a, H-14b), 1.97 (brs, 1H, OH-1), 1.65 (d, *J* =1.7 Hz, 3H, CH_3_), 1.23 (s, 3H, CH_3_), 1.16 (s, 3H, CH_3_); ^13^C-NMR (CDCl_3_, 100 MHz) δ: 191.5, 172.4, 169.7, 167.0, 165.5, 153.8, 152.6, 144.5, 138.0, 135.9, 134.8, 133.8, 133.7, 132.0, 130.9, 130.2 (2C), 129.0, 128.7, 128.4, 127.1, 127.0, 90.4, 84.6, 78.5, 76.1, 75.5, 73.2, 71.7, 69.2, 65.9, 64.4, 55.0, 42.8, 42.7, 36.4, 30.9, 29.0, 26.1, 22.3, 21.2, 14.6, 14.0; LC-MS (ESI, *m/z*): [M + 1], found 930.3, [M + 23], found 952.3, C_49_H_52_ClNO_15_: 929.30. 

*2-Debenzoyl-2-(3-trifluoromethylbenzoyl)-macrocyclic taxoid* (**22e**). White solid; yield 59.5%; ^1^H-NMR (CDCl_3_, 400 MHz) δ: 8.46 (s, 1 H, Ar), 8.36 (d, *J* = 7.8 Hz, 1H, Ar-H), 7.90 (d, *J* =7.7 Hz, 1H, Ar-H), 7.73–7.67 (m, 3 H, Ar-H), 7.51–7.47 (m, 3H, Ar-H), 7.42–7.32 (m, 5H, Ar-H), 6.98 (d, *J* = 9.0 Hz, 1H, NH), 6.15 (t, *J* = 8.1 Hz, 1H, H-13), 5.77 (dd, *J* = 2.0, 9.0 Hz, 1H, H-3′), 5.61 (d, *J* = 7.7 Hz, 1H, H-2), 4.84–4.78 (overlap, 3H, H-5, H-3, H-2′), 4.62–4.51 (m, 2H, OCH_2_CH_2_CH_2_OCO), 4.45–4.41 (m, 2H, H-7a, OCH_2_CH_2_CH_2_OCO), 4.39 (d, *J* = 8.0 Hz, 1H, H-20a), 4.31 (d, *J* = 8.2 Hz, 1H, H-20b), 4.07 (t, *J* = 7.6 Hz, 1H, OCH_2_CH_2_CH_2_OCO), 3.93 (dt, *J* = 6.9, 11.6 Hz, 1H, H-7b), 3.62 (brs, 1H, OH-2′), 2.66–2.58 (m, 1H, H-6a), 2.54 (s, 3H, CH_3_), 2.44–2.35 (m, 2H, OCH_2_CH_2_CH_2_OCO), 2.32 (s, 3H, CH_3_), 2.28–2.16 (m, 3H, H-6b, H-14a, H-14b), 1.97 (brs, 1H, OH-1), 1.67 (br, 3H, CH_3_), 1.23 (s, 3H, CH_3_), 1.17 (s, 3H, CH_3_); ^13^C-NMR (CDCl_3_, 75 MHz) δ: 191.6, 172.4, 169.7, 167.0, 165.4, 153.7, 152.6, 144.5, 138.1 (2C), 135.8, 133.7, 133.6, 131.9, 131.3 (d, ^2^*J*_C-F_ = 33.2 Hz), 130.3 (d, ^3^*J*_C-F_ = 3.7 Hz), 130.1, 129.6, 129.0, 128.7, 128.4, 127.1, 127.0, 126.9 (d, ^3^*J*_C-F_ = 3.2 Hz), 125.5, 90.5, 84.6, 78.5, 76.1, 75.6, 73.1, 71.7, 69.2, 66.0, 64.4, 55.0, 42.8, 42.7, 36.3, 30.9, 29.3, 28.9, 26.1, 22.2, 21.2, 14.6, 14.0; LC-MS (ESI, *m/z*): [M + 1], found 964.4, [M + 23], found 986.3, C_50_H_52_F_3_NO_15_: 963.33. 

### 3.2. Cell Assays

Cell viability after taxoids treatment was evaluated using CCK-8 assay. Briefly, 3000 cells per well were seeded in 96-well plates and incubated under normal conditions for 24 h. Cells were treated with different concentrations of the test agent for 72 h, then the medium was removed, 100 μL of CCK-8 working solution was added to each well for 1 h at 37 °C. The absorbance was measured at 450 nm with a microplate reader (Tecan Trading, AG, Switzerland). Vehicle-only treated cells served as the indicator of 100% cell viability. The 50% inhibitory concentration (IC_50_) was defined as the concentration that reduced the absorbance of the vehicle-only treated wells by 50% in the 3-(4,5-dimethylthiazol-2-yl)-2,5-diphenyltetrazolium bromide (MTT) assay.

### 3.3. Model Building and Molecular Dynamics Simulations

The X-ray crystal structure of paclitaxel provided a template for the construction of **18b** and **22b** using the molecular editing tools implemented in PyMOL [17]. Geometry optimization of **18b** and **22b** was achieved by means of the updated AM1 hamiltonian implemented in the *sqm* program, which also produced atomic charge distributions for both ligands that can reproduce the molecular electrostatic potential. The ff14SB AMBER force field was used to assign bonded and non-bonded parameters to protein and ligand atoms [18,19]. **18b** was immersed in a cubic box containing TIP3P water molecules and simulated under periodic boundary conditions for 50 ns at 300 K. Subsequent gradual cooling, from 300 to 273 K over 1 ns, of snapshots taken regularly every 2.5 ns, followed by energy minimization until the root-mean-square of the Cartesian elements of the gradient was less than 0.1 kcal·mol^−1^·Å^−1^, provided representative structures of this molecule; those displaying the shortest distance between the free hydroxyl groups were chosen to build **22b** by means of a C=O linkage. The simulated macromolecular ensemble representing a short piece of a microtubule with bound **22b** was constructed as previously reported for D-seco taxol derivatives [20].

## 4. Conclusions

In summary, this study demonstrated that C-seco taxoids conformationally constrained via carbonate-containing linked macrocyclization display increased cytotoxicity on drug-resistant tumors overexpressing both βIII and P-gp, compared to the non-cyclized C-seco taxoid counterparts Among them, compound **22b**, bearing a 2-*m*-methoxybenzoyl group together with a five-atom linker, was identified as the most potent compound in the series.

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
