# Peer review of "Synthesis and Cytotoxicity of 7,9-*O*-Linked Macrocyclic C-Seco Taxoids"

_molecules, 2019, doi:10.3390/molecules24112161_

Round 1
Reviewer 1 Report
The authors synthesize several macrocyclic taxoids to study their citotoxicity against drug resistant tumors. They evaluate the compounds against cancer cell lines and resistant cancer cell lines and show how the macrocyclic taxoids do show and increased activity against resistant cell lines compared to non cyclized derivatives.
The manuscript is clear and well written.
I just have some minor questions or comments to improve the manuscript.
The macorcyclic taxoids show higher citoxocity that the non cyclized derivatives but very similar to paclitaxel, would they have any advantage to paclitaxel? The binding mechanism showed with the docking studies would be the same for paclitaxel?, Please comment on this on the discussion.
Please change scheme 1 to make structures bigger. Why is the synthesis of 10, 11 and 12 if it is not performed/described here? If removed the structures could be made bigger.
I did not find the HPLC data of the compounds,the gradient used and retention time for the compounds should be added to the methods.
I found some OH signals were described for some compounds but not for all. Please check the assignments.
Author Response
June 5, 2019
Dear reviewer,
Thank you for your effort to improve the quality of our manuscript. Please find in the following pages our replies to your’ comments.
We wish these revisions meet your expectations and make this manuscript publishable in its current form.
Looking forward to hearing positive feedback from you!
Dr. Weishuo Fang
Professor of Medicinal Chemistry
Institute of Materia Medica, CAMS
2A Nan Wei Road, Beijing 100050
China
Email: wfang@imm.ac.cn
Tel: +86(10)63165229
Web: http://www.imm.ac.cn/groups/fangws/
Response to Reviewer 1 Comments
The authors synthesize several macrocyclic taxoids to study their citotoxicity against drug resistant tumors. They evaluate the compounds against cancer cell lines and resistant cancer cell lines and show how the macrocyclic taxoids do show and increased activity against resistant cell lines compared to non cyclized derivatives.
The manuscript is clear and well written.
I just have some minor questions or comments to improve the manuscript.
Point 1: The macorcyclic taxoids show higher citoxocity that the non cyclized derivatives but very similar to paclitaxel, would they have any advantage to paclitaxel? The binding mechanism showed with the docking studies would be the same for paclitaxel?, Please comment on this on the discussion.
Response 1: We have add some words in section 2.3 to discuss it.
As shown in this modelling study, the binding of these taxoids to microtubules are similar to that of paclitaxel. Although cyclization does increase the cytotoxicity either in drug sensitive and resistant cells in most cases, the 7,9-O-linked macrocyclic taxoids only showed comparable cytotoxicity to that of paclitaxel. The enhanced activity is not only arisen from the known C-2 modifications, but also from the cyclization of the cleaved C-ring of taxane.
Point 2: Please change scheme 1 to make structures bigger. Why is the synthesis of 10, 11 and 12 if it is not performed/described here? If removed the structures could be made bigger.
Response 2: The scheme and the size for structures have been adjusted accordingly.
Point 3: I did not find the HPLC data of the compounds, the gradient used and retention time for the compounds should be added to the methods.
Response 3: HPLC conditions have been added to the experimental section.
All tested compounds 16a-e, 21a-e and 22a-e were ≥ 95% pure by HPLC (column XDB C18 4.6x50 mm 1.8 µm, mobile phase: acetonitrile-water (10 : 90 – 100 : 0 gradient in 4.5 min), flow rate 1.0 mL/min, detected at 220 nm).
Point 4: I found some OH signals were described for some compounds but not for all. Please check the assignments.
Response 4: Only a part of OH signals are visible in the proton NMR, probably because some more active OH signals have been exchanged by the moisture in either sample or solvent.
Reviewer 2 Report
This manuscript by Dr. Fang and co-workers describes the synthesis and cytotoxicity of macrocyclic taxoids.
Taxol and its derivatives are well known to be highly valuable for treating several types of cancers, however, drug resistance are considered problematic in their clinical usages.
The authors reported that 7,9-O-linked C-seco taxoid showed potent activity against Hela-βIII in their previous report. In related to the previous work, they prepared several macrocylic compounds and C-seco toxoids and biologically evaluated fifteen synthetic compounds. This study revealed that incorporating a carbonate in the macrocycle and substitution at the meta-position of the C2 benzoyl moiety were highly important to exhibit potent cytotoxicity. Among the synthetic compounds, 22b was found to show the most potent activity, which could be explained by plausible effective interaction with the T7 loop region of βIII. These findings should stimulate interest to pharmaceutical chemists.
Thus, the reviewer thinks this manuscript is suitable to be published in molecules after minor revision as pointed out below.
Comments:
(a) Abbreviations (such as MDR) should be defined at first mention.
(b) Figures 1 and 3: The number in the R group should be superscript.
(c) Lines 15 and 103, and line 3 in page 8: Is “carbonate-mediated macrocyclization” suitable term? Because, although the macrocycle contains a carbonate, macrocyclization is not mediated by a carbonate.
(d) Line 85: NH2NH2.H2O should be NH2NH2·H2O.
(e) Line 118: A space between “here” and “.” should be deleted.
(f) Line 135: A period should be added at the end of the text.
(g) Experimental section and References: There are some typing errors. Please check the text carefully again.
Author Response
June 5, 2019
Dear reviewer,
Thank you for your effort to improve the quality of our manuscript. Please find in the following pages our replies to your’ comments.
We wish these revisions meet your expectations and make this manuscript publishable in its current form.
Looking forward to hearing positive feedback from you!
Dr. Weishuo Fang
Professor of Medicinal Chemistry
Institute of Materia Medica, CAMS
2A Nan Wei Road, Beijing 100050
China
Email: wfang@imm.ac.cn
Tel: +86(10)63165229
Web: http://www.imm.ac.cn/groups/fangws/
Response to Reviewer 2 Comments
This manuscript by Dr. Fang and co-workers describes the synthesis and cytotoxicity of macrocyclic taxoids.
Taxol and its derivatives are well known to be highly valuable for treating several types of cancers, however, drug resistance are considered problematic in their clinical usages.
The authors reported that 7,9-O-linked C-seco taxoid showed potent activity against Hela-βIII in their previous report. In related to the previous work, they prepared several macrocylic compounds and C-seco toxoids and biologically evaluated fifteen synthetic compounds. This study revealed that incorporating a carbonate in the macrocycle and substitution at the meta-position of the C2 benzoyl moiety were highly important to exhibit potent cytotoxicity. Among the synthetic compounds, 22b was found to show the most potent activity, which could be explained by plausible effective interaction with the T7 loop region of βIII. These findings should stimulate interest to pharmaceutical chemists.
Thus, the reviewer thinks this manuscript is suitable to be published in molecules after minor revision as pointed out below.
Comments:
Point 1: Abbreviations (such as MDR) should be defined at first mention.
Response 1: Done.
Point 2: Figures 1 and 3: The number in the R group should be superscript.
Response 2: Revised accordingly.
Point 3: Lines 15 and 103, and line 3 in page 8: Is “carbonate-mediated macrocyclization” suitable term? Because, although the macrocycle contains a carbonate, macrocyclization is not mediated by a carbonate.
Response 3: “carbonate-mediated macrocyclization” have been changed to “carbonate containing-linked macrocyclization”.
Point 4: Line 85: NH2NH2.H2O should be NH2NH2·H2O.
Response 4: Revised.
Point 5: Line 118: A space between “here” and “.” should be deleted.
Response 5: Done.
Point 6: Line 135: A period should be added at the end of the text.
Response 6: Done.
Point 7: Experimental section and References: There are some typing errors. Please check the text carefully again.
Response 7: Revised accordingly.
Reviewer 3 Report
This paper is a valuable contribution that explores the Synthesis and cytotoxicity of 9,7-O-linked macrocyclic C-seco taxoids and their cytotoxicities against drug-sensitive and P-glycoprotein and 14 βIII-tubulin overexpressed drug resistant cancer cell lines; several techniques were explored and contains a great deal of information. However, I cannot recommend this manuscript to be published in this version in molecules due to the issues as follows.
Scheme 1. The abbreviation of tert-Butyldimethylsilyl ethers expressed in the reactions most used TBDMS an not as TBS
In my opinion all the yield could be better if are written next to the number of the compound for example 11a R= O-CH3; 81.8% because in the text of the scheme is very difficult to see the reaction conditions.
Line 25 if all the compound were purified by HLPC why do not were separate the epimers.
In the experimental section are several mistakes for example: line 82 CH3 and could be CH3
The characterization of the compound 14a is incomplete.
The number of protons in the elemental analysis is different to the number in the text for the compound 14b, 14d. Please check the protons for all the compounds.
Author Response
June 5, 2019
Dear reviewer,
Thank you for your effort to improve the quality of our manuscript. Please find in the following pages our replies to your’ comments.
We wish these revisions meet your expectations and make this manuscript publishable in its current form.
Looking forward to hearing positive feedback from you!
Dr. Weishuo Fang
Professor of Medicinal Chemistry
Institute of Materia Medica, CAMS
2A Nan Wei Road, Beijing 100050
China
Email: wfang@imm.ac.cn
Tel: +86(10)63165229
Web: http://www.imm.ac.cn/groups/fangws/
Response to Reviewer 3 Comments
This paper is a valuable contribution that explores the Synthesis and cytotoxicity of 9,7-O-linked macrocyclic C-seco taxoids and their cytotoxicities against drug-sensitive and P-glycoprotein and 14 βIII-tubulin overexpressed drug resistant cancer cell lines; several techniques were explored and contains a great deal of information. However, I cannot recommend this manuscript to be published in this version in molecules due to the issues as follows.
Point 1: Scheme 1. The abbreviation of tert-Butyldimethylsilyl ethers expressed in the reactions most used TBDMS an not as TBS.
Response 1: Done.
Point 2: In my opinion all the yield could be better if are written next to the number of the compound for example 11a R= O-CH3; 81.8% because in the text of the scheme is very difficult to see the reaction conditions.
Response 2: Revised as suggested by the reviewer.
Point 3: Line 25 if all the compound were purified by HLPC why do not were separate the epimers.
Response 3: The epimer mixture of 7-epi-10-dehydro-10-deacetylpaclitaxel 14 will be treated with NaBH4 and CeCl3•7H2O next to furnish a homogeneous product 15. This transformation is a followup of known reactions (refer to Tetrahedron Lett., 1995, 36, 3233-3236). So we chose not to separate the epimers which are subject to the next step transformation directly.
Point 4: In the experimental section are several mistakes for example: line 82 CH3 and could be CH3
Response 4: Revised.
Point 5: The characterization of the compound 14a is incomplete.
Response 5: Compound 14a is a known compound and its chemical characterization can be found in the literature (European Journal of Medicinal Chemistry (2017), 137, 488-503. DOI:10.1016/j.ejmech.2017.06.001). The 1H-NMR spectra of the two compounds are identical.
Point 6: The number of protons in the elemental analysis is different to the number in the text for the compound 14b, 14d. Please check the protons for all the compounds.
Response 6: The 1-OH signals in 14b and 14d disappeared in the proton NMR, due to deuterium exchange by moisture, thus made the number of protons in the elemental analysis is different from that in NMR.